

# $q$th-root non-Hermitian Floquet topological insulators

**Longwen Zhou[1]⋆, Raditya Weda Bomantara[2]† and Shenlin Wu[1]**

**1** College of Physics and Optoelectronic Engineering, Ocean University of China,
Qingdao 266100, China
**2** Centre for Engineered Quantum Systems, School of Physics, University of Sydney,
Sydney, New South Wales 2006, Australia

⋆ zhoulw13@u.nus.edu, † Raditya.Bomantara@sydney.edu.au

## Abstract

Floquet phases of matter have attracted great attention due to their dynamical and topological nature that are unique to nonequilibrium settings. In this work, we introduce a generic way of taking any integer $q$th-root of the evolution operator $U$ that describes Floquet topological matter. We further apply our $q$th-rooting procedure to obtain $2^n$th- and $3^n$th-root first- and second-order non-Hermitian Floquet topological insulators (FTIs). There, we explicitly demonstrate the presence of multiple edge and corner modes at fractional quasienergies $\pm(0, 1, ...2^n)\pi/2^n$ and $\pm(0, 1, ..., 3^n)\pi/3^n$, whose numbers are highly controllable and capturable by the topological invariants of their parent systems. Notably, we observe non-Hermiticity induced fractional-quasienergy corner modes and the coexistence of non-Hermitian skin effect with fractional-quasienergy edge states. Our findings thus establish a framework of constructing an intriguing class of topological matter in Floquet open systems.



# 1  Introduction

Periodically driven (Floquet) systems have attracted perennial interest owing to their fascinating dynamical, topological and transport properties (see Refs. [1–6] for reviews). Theoretical classifications of Floquet matter have been achieved for both free [7–9] and interacting [10–12] systems. Experimental observations of Floquet phases have also been made in cold atoms [13–15], photonics [16–18] and solid state materials [19–21], boosting the developments of new ideas in ultrafast electronics [4] and topological quantum computing [22–24].

Recently, square-root topological phase is discovered [25], whose topological properties are inherited from its squared parent model through a process analogous to the transition from Klein-Gordon [26,27] to Dirac equations [28] in relativistic quantum mechanics. In-gap edge modes are found in tight-binding models of square-root topological insulators, superconductors and semimetals [29–41]. Moreover, general rules of constructing $2^n$th-root topological phases [35–37] and their symmetry classifications [38] are proposed. Experimental evidence of square-root topological phases are reported in photonic [30], electric [31] and acoustic [32] systems.

In a periodically driven system, the central object for the description of topological properties is the Floquet operator, which is the evolution operator of the system over a complete driving period $T$. Taking the square-root of such a propagator for the purpose of generating its topological descendant is, however, a highly nontrivial task. This can be seen by writing the Floquet operator as $U = \mathcal{T} e^{-\frac{i}{\hbar} \int_0^T H(t) dt} = e^{-i \frac{T}{\hbar} H_{\text{eff}}}$, where $\mathcal{T}$ is the time-ordering operator, $H(t) = H(t + T)$ is the time-periodic Hamiltonian of the system, and $H_{\text{eff}}$ is the Floquet effective Hamiltonian obtained by formally working out the time-ordered product in $\mathcal{T} e^{-\frac{i}{\hbar} \int_0^T H(t) dt}$. We may now take the square-root of $U$ naively as $\sqrt{U} = e^{-i \frac{T}{\hbar} \frac{H_{\text{eff}}}{2}}$. However, such a trial of generating square-root Floquet topological phases tends out to be problematic and useless. First, the exact form of $H_{\text{eff}}$ can be rather complicated (usually including driving-induced long-range coupling terms), not physically obtainable, or even insufficient to describe Floquet phases with no static counterparts such as those possessing anomalous Floquet edge modes [42–44]. Second, there are no transparent ways to find $H_{\text{eff}}$ from $H(t)$, i.e., the parameters in $H_{\text{eff}}$ are usually nonlinear combinations of physical parameters in $H(t)$, such that simply reducing the parameters of $H(t)$ by half could not yield $H_{\text{eff}}/2$. Even obtained, the $H_{\text{eff}}$ and $H_{\text{eff}}/2$ describe essentially the same physical system up to a global constant, and no new physics are expected to emerge following such a halving process. Therefore, the straightforward operation, $\sqrt{U} = e^{-i \frac{T}{\hbar} \frac{H_{\text{eff}}}{2}}$, does not generate a desired square-root of the parent system $U$.

To resolve this puzzle, a nontrivial route of taking the square-root for $U$ is introduced [45], which closely follows the original idea of Dirac by adding internal degrees of freedom for electrons before taking the square-root of their relativistic wave equation. However, the general applicability of this idea to the construction of Floquet models beyond taking $2^n$th-root has not been revealed. Motivated by this gap of knowledge, we propose a generic procedure to yield a variety of $q$th-root Floquet phases, where $q$ is *any* arbitrary integer, not necessarily in the form of $2^n$. This is achieved by utilizing a $Z_q$ generalization of Pauli matrices as ancillary degrees of freedom. While our construction is applicable to any periodically driven systems,

we focus on two timely examples of non-Hermitian Floquet matter as case studies.

The concept of topological matter has been generalized to non-Hermitian systems in recent years (see Refs. [46–50] for reviews). In the presence of gain and loss or nonreciprocal effects, unique topological phenomena without any counterparts in closed systems could emerge, such as the non-Hermitian skin effect (NHSE) [51–57] and exceptional topological phases [50]. The interplay between time-periodic drivings and non-Hermitian effects could further induce intriguing phases in out-of-equilibrium situations, like the non-Hermitian Floquet topological insulators [58–69], superconductors [70,71], semimetals [72–75] and quasicrystals [76–78]. As reported in this paper, applying our $q$th-rooting procedure to such non-Hermitian Floquet phases yields even more exotic features absent in their original counterparts, such as fractional-quasienergy topological edge and corner modes.

This paper is structured as follows. In Sec. 2, we recap the strategy of Ref. [45], generalize it to the construction of *any* $q$th-root Floquet system, and elaborate the application of this general construction for the case of $q = 3$. In Sec. 3, we introduce two typical models of first- and second-order non-Hermitian Floquet topological insulators, whose square- and cubic-root descendants are studied in detail in Sec. 4 as an application of our method. In Sec. 5, we sum up our results and discuss potential future directions.

## 2 Theory

We first review the approach to take the nontrivial square-root of a Floquet system. We set the Planck constant $\hbar = 1$ and driving period $T = 1$ throughout. Following Ref. [45], we write the one-period evolution (Floquet) operator of *any* time-periodic system as

$$U = U_1 U_2 = \left( \mathcal{T} e^{-i \int_0^{1/2} H(t+1/2) dt} \right) \left( \mathcal{T} e^{-i \int_0^{1/2} H(t) dt} \right), \tag{1}$$

where $H(t)$ is the system's Hamiltonian. The procedure of Ref. [45] is to first enlarge Hilbert space of $H(t)$ by introducing a pseudospin-1/2 degree of freedom with the corresponding Pauli matrices $\tau_{x,y,z}$. A two-step Hamiltonian is next defined in the enlarged Hilbert space as

$$H_{1/2}(t) = \begin{cases} \pi \tau_y \otimes \mathbb{I}_0, & t \in [\ell, \ell + \frac{1}{2}) \\ \frac{\tau_0 + \tau_z}{2} \otimes H(t) + \frac{\tau_0 - \tau_z}{2} \otimes H(t+1/2), & t \in [\ell + \frac{1}{2}, \ell + 1) \end{cases}, \tag{2}$$

where $\ell \in \mathbb{Z}$. $\tau_0$ is the identity in the pseudospin-1/2 subspace. $\mathbb{I}_0$ is the identity in the Hilbert space of $H(t)$. The Floquet operator of the evolution in the enlarged Hilbert space reads

$$U_{1/2} = \begin{pmatrix} \mathcal{T} e^{-i \int_{1/2}^1 H(t) dt} & 0 \\ 0 & \mathcal{T} e^{-i \int_{1/2}^1 H(t+1/2) dt} \end{pmatrix} e^{-i \frac{\pi}{2} \tau_y \otimes \mathbb{I}_0}. \tag{3}$$

Note that $\mathcal{T} e^{-i \int_{1/2}^1 H(t) dt} = \mathcal{T} e^{-i \int_0^{1/2} H(t+1/2) dt} = U_1$ and $\mathcal{T} e^{-i \int_{1/2}^1 H(t+1/2) dt} = \mathcal{T} e^{-i \int_0^{1/2} H(t) dt} = U_2$. Performing the Taylor expansion and introducing $\tau_\pm = (\tau_x \pm i \tau_y)/2$, we find

$$U_{1/2} = \tau_- \otimes U_2 - \tau_+ \otimes U_1, \tag{4}$$

and

$$U_{1/2}^2 = e^{i\pi} \begin{pmatrix} U_1 U_2 & 0 \\ 0 & U_2 U_1 \end{pmatrix}. \tag{5}$$

Since $U_1 U_2 = U$ and $U_2 U_1 = U_2 U U_2^{-1}$ are related by a similarity transformation, they describe the same parent Floquet system up to a half-period shift of the initial evolution time. The

system described by $U_{1/2}^2$ can thus be viewed as two equivalent copies of $U$ up to a global phase shift $\pi$. Therefore, $U_{1/2}^2$ and $U$ are expected to share the same topological features concerning the stroboscopic dynamics. We could view $U_{1/2}$ as a nontrivial square-root of $U$ in the spirit of Dirac's taking square-root to reach his equation for electrons [28]. The dynamical and topological properties of $U$ can further be carried over to $U_{1/2}$, which are confirmed by explicit studies of Floquet topological superconductors and time crystals [45].

Iterating the same procedure, we can construct the $2^n$th-root of $U$, i.e., $U_{1/2^n}$ for any $n \in \mathbb{Z}^+$. For example, we could generate $U_{1/4}$ by letting $H_1'(t) = \frac{\tau_0 + \tau_z}{2} \otimes H(t) + \frac{\tau_0 - \tau_z}{2} \otimes H(t + \frac{1}{2})$, $H_2' = \pi \tau_y \otimes \mathbb{I}_0$, and defining in a further enlarged Hilbert space

$$H_{1/4}(t) = \begin{cases} \pi \tau_y' \otimes \mathbb{I}_0', & t \in [\ell, \ell + \frac{1}{2}) \\ \frac{\tau_0' + \tau_z'}{2} \otimes H_1'(t) + \frac{\tau_0' - \tau_z'}{2} \otimes H_2', & t \in [\ell + \frac{1}{2}, \ell + 1) \end{cases}, \tag{6}$$

where $\tau_{y,z}'$ and $\tau_0'$ are Pauli matrices and identity matrix acting in the subspace of an extra pseudospin-1/2. $\mathbb{I}_0'$ denotes the identity in the Hilbert space of $H_{1,2}'$. The resulting Floquet operator,

$$U_{1/4} = \left( \mathcal{T} e^{-i \int_{1/2}^{1} \left( \frac{\tau_0' + \tau_z'}{2} \otimes H_1'(t) + \frac{\tau_0' - \tau_z'}{2} \otimes H_2' \right) dt} \right) e^{-i \frac{\pi}{2} \tau_y' \otimes \mathbb{I}_0'}, \tag{7}$$

then defines the nontrivial 4th-root of $U$. It is straightforward to verify that

$$U_{1/4}^4 = e^{i\pi} \begin{pmatrix} U_1 U_2 & 0 & 0 & 0 \\ 0 & U_2 U_1 & 0 & 0 \\ 0 & 0 & U_2 U_1 & 0 \\ 0 & 0 & 0 & U_1 U_2 \end{pmatrix}, \tag{8}$$

whose diagonal blocks contain four equivalent copies of $U$ up to a unitary transformation and a global phase $\pi$.

The extension of the above approach to find any $q$th-root of $U$ can be achieved by introducing higher-dimensional pseudospin degrees of freedom, i.e., the generalized $q \times q$ Pauli matrices

$$[\eta_x]_{i,j} = \delta_{i,j-1} + \delta_{i,q} \delta_{j,1}, \qquad [\eta_z]_{i,j} = \omega^{j-1} \delta_{i,j}, \tag{9}$$

where $\omega = e^{i2\pi/q}$. These operators satisfy

$$\eta_x \eta_z = \omega \eta_z \eta_x, \quad \eta_x \eta_x^\dagger = \eta_z \eta_z^\dagger = \eta_0, \quad \eta_x^q = \eta_z^q = \eta_0, \tag{10}$$

where $\eta_0$ is the identity matrix acting in the pseudospin subspace. Our $q$th-rooting procedure can then be executed in two steps. First, given any time-periodic Hamiltonian $H(t)$, we divide the Floquet operator into $q$ time-steps, i.e.,

$$U = \prod_{\ell=1}^{q} U_\ell = U_1 \cdots U_q, \quad U_\ell = \mathcal{T} e^{-i \int_0^{1/q} H\left(t + \frac{q-\ell}{q}\right) dt}. \tag{11}$$

Next, we define a two-step Hamiltonian of the form

$$H_{1/q}(t) = \begin{cases} \sum_{j=0}^{q-1} M_j^{(q)} \eta_x^j \otimes \mathbb{I}_0, & t \in [\ell, \ell + \frac{1}{2}) \\ \sum_{j=1}^{q} \mathcal{P}_j^{(q)} \otimes \tilde{H}_j(t), & t \in [\ell + \frac{1}{2}, \ell + 1) \end{cases}, \tag{12}$$

where $M_0^{(q)} = M_0^{(q)\dagger}$ and $M_{j\neq0}^{(q)} = M_{q-j}^{(q)\dagger}$ for hermiticity. $M_j^{(q)}$ are further chosen such that $e^{-i\sum_{j=0}^{q-1} M_j^{(q)} \eta_x^j} = \eta_x$. $\mathbb{I}_0$ is the identity in the Hilbert space of $H(t)$, $\mathcal{P}_j^{(q)} = \frac{\sum_{k=0}^{q-1} \omega^{k(j-1)} \eta_z^k}{q}$, and $\tilde{H}_j(t) = \frac{2H\left(2(t+\frac{q-j}{q})/q\right)}{q}$. It then follows that the associated Floquet operator is

$$[U_{1/q}]_{i,j} = U_i \delta_{i,j-1} + U_q \delta_{i,q} \delta_{j,1}, \tag{13}$$

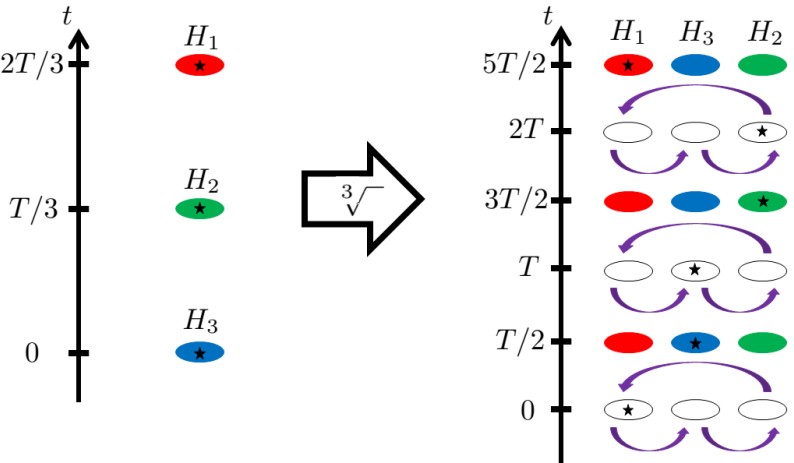

Figure 1: Schematic of a cubic-root system obtained from the procedure of Eq. (12). In the parent system, a particle (marked by black star) evolves under $H_3$, $H_2$, and $H_1$ over the course of a period. In the corresponding cubic-root system, a particle living in a given subsystem effectively evolves under the same three Hamiltonians only when viewed over the course of three periods.

such that $U_{1/q}^q$ is block diagonal and consists of all $q$ permutations of $\prod_{\ell=1}^q U_\ell$. This shows that the resulting system indeed represents the $q$th-rooted version of $U$.

Intuitively, the above construction can be understood as follows. First, note that Eq. (12) is defined on a system consisting of $n$ subsystems. Consider a particle initially living in the $j$th subsystem. During the first half of the period, evolution under Eq. (12) amounts to transporting the particle towards subsystem $j-1 \mod q$. By noting that $\mathcal{P}_j^{(q)}$ represents a projection onto the $j$th subsystem, it then follows that Eq. (12) further evolves the particle under $H_{j-1}(t)$ during the second half of the period. In the next Floquet cycle, the particle continues moving to subsystem $j-2 \mod q$, followed by the half-period evolution under $H_{j-2}(t)$. As the process continues, at the end of $q$ periods, the particle returns to the subsystem $j \mod q$, while accumulating $U_j \cdots U_q U_1 \cdots U_{j-2} U_{j-1}$, which is unitarily equivalent to $U$.

Having demonstrated the generality of our construction, we will focus on square-root and cubic-root systems for brevity in the remainder of this paper. To this end, we will now present an explicit application of the above construction to obtain a nontrivial cubic-root of a system relevant to the case studies below. Specifically, such a parent system follows a three-step periodically quenched drive, whose time-dependent Hamiltonian takes the form

$$H(t) = \begin{cases} H_1, & t \in [\ell + 2/3, \ell + 1) \\ H_2, & t \in [\ell + 1/3, \ell + 2/3) \qquad \ell \in \mathbb{Z}, \\ H_3, & t \in [\ell, \ell + 1/3) \end{cases} \tag{14}$$

with the corresponding Floquet operator of

$$U = e^{-i\frac{H_1}{3}} e^{-i\frac{H_2}{3}} e^{-i\frac{H_3}{3}}. \tag{15}$$

Note that the Floquet operator associated with a system following a two-step periodically quenched drive, whose Hamiltonian switches between $h_1$ and $h_2$ after every half period, can also be cast into the form of Eq. (15) by shifting the initial evolution time from $t = 0$ to $3/4$ and identifying $H_1, H_3 = 3h_1/4$ and $H_2 = 3h_2/2$. In both cases, the cubic-root of $U$ in Eq. (15) can be obtained according to Eq. (12) with $M_0^{(3)} = 0$ and $M_2^{(3)} = -M_1^{(3)} = \frac{4\pi i}{3\sqrt{3}}$, which leads to

$e^{-i\left(M_1^{(3)}\eta_x + M_2^{(3)}\eta_x^2\right)} = \eta_x$. We further identify $\tilde{H}_j = \frac{2H_{3-j}}{3}$ with $j = 0, 1, 2$, as well as $\eta_{x,z}$ in the explicit matrix forms

$$\eta_x = \begin{pmatrix} 0 & 1 & 0 \\ 0 & 0 & 1 \\ 1 & 0 & 0 \end{pmatrix}, \qquad \eta_z = \begin{pmatrix} 1 & 0 & 0 \\ 0 & \omega & 0 \\ 0 & 0 & \omega^2 \end{pmatrix}. \tag{16}$$

Indeed, it can be directly verified that Eq. (16) satisfies the algebra of Eq. (10). The corresponding Floquet operator of the cubic-root model then reads

$$U_{1/3} = \begin{pmatrix} 0 & e^{-i\frac{\tilde{H}_1}{3}} & 0 \\ 0 & 0 & e^{-i\frac{\tilde{H}_2}{3}} \\ e^{-i\frac{\tilde{H}_3}{3}} & 0 & 0 \end{pmatrix}, \tag{17}$$

with

$$U_{1/3}^3 = \begin{pmatrix} e^{-i\frac{\tilde{H}_1}{3}}e^{-i\frac{\tilde{H}_2}{3}}e^{-i\frac{\tilde{H}_3}{3}} & 0 & 0 \\ 0 & e^{-i\frac{\tilde{H}_2}{3}}e^{-i\frac{\tilde{H}_3}{3}}e^{-i\frac{\tilde{H}_1}{3}} & 0 \\ 0 & 0 & e^{-i\frac{\tilde{H}_3}{3}}e^{-i\frac{\tilde{H}_1}{3}}e^{-i\frac{\tilde{H}_2}{3}} \end{pmatrix}. \tag{18}$$

That is, the three diagonal blocks of $U_{1/3}^3$ only differ from one another by the starting time of evolution and describe equivalent Floquet systems in stroboscopic dynamics concerning the spectral and topological properties. This implies that $U_{1/3}$ is indeed a nontrivial cubic-root of the parent system $U$ in Eq. (15). The presented cubic root procedure is schematically depicted in Fig. 1.

We now discuss how the rooted Floquet system could inherit the symmetry protected edge states of the parent model while altering their quasienergies to rational fractions of $2\pi$. A key symmetry that is relevant to the topological characterization of the parent systems considered in this work is the chiral symmetry (CS). If a general Floquet operator $U$ possesses the CS, there is a unitary operator $\Gamma$ such that $\Gamma U \Gamma^\dagger = U^{-1}$. If $U$ has an eigenstate $|\psi\rangle$ with quasienergy $E$, i.e., $U|\psi\rangle = e^{-iE}|\psi\rangle$, its CS implies that $\Gamma U \Gamma^\dagger(\Gamma|\psi\rangle) = e^{-iE}(\Gamma|\psi\rangle)$ or $U(\Gamma|\psi\rangle) = e^{-i(-E)}(\Gamma|\psi\rangle)$. Therefore, $\Gamma|\psi\rangle$ is an eigenstate of $U$ with quasienergy $-E$. Now if there is an eigenstate $|\psi\rangle$ with $E = 0$ or $\pi$, the CS enforces the presence of another eigenstate $\Gamma|\psi\rangle$ also at $E = 0$ or $\pi$ ($E = \pm\pi$ are identified as the same quasienergy since $E$ is defined mod $2\pi$), yielding eigenstate degeneracy at the center or boundary of the quasienergy Brillouin zone $E \in [-\pi, \pi]$. If such eigenmodes appear at the edge or corner of the system, we obtain CS-protected degenerate edge or corner modes of $U$.

For the square-root system $U_{1/2}$, we already see that $U_{1/2}^2$ is block diagonal and its two diagonal blocks share the same spectral and topological properties with the parent model $U$. If $|\psi'\rangle$ is an eigenstate of $U_{1/2}$ with quasienergy $E'$, i.e., $U_{1/2}|\psi'\rangle = e^{-iE'}|\psi'\rangle$, it is straightforward to see that $U_{1/2}^2|\psi'\rangle = e^{-iE'}U_{1/2}|\psi'\rangle = e^{-i2E'}|\psi'\rangle$. $U_{1/2}$ and $U_{1/2}^2$ thus share the same eigenbasis. When the parent model $U$ possesses the CS $\Gamma$, the diagonal blocks of $U_{1/2}^2$ possess the CS, such that $U_{1/2}^2$ is chiral symmetric with respect to $\Gamma' = \tau_z \otimes \Gamma$. Degenerate topological edge/corner modes of $U_{1/2}^2$ can thus only appear at $E = 0, \pm\pi = 2E'$ mod $2\pi$. This implies that in the square-root system $U_{1/2}$, we could only find topological edge/corner modes at the quasienergies $E' = 0, \pm\pi/2, \pm\pi$, which are indeed protected by the CS $\Gamma$ of the parent model. Interestingly, the degenerate eigenmodes at $E = \pm\pi/2$ are present only in the $U_{1/2}$ and are thus unique to the square-root Floquet system.

While the topological protection of the edge/corner modes in $U_{1/2}$ can be understood from the presence of chiral symmetry in its corresponding parent system $U$, it would also be insightful to discuss the protecting symmetries that arise at the level of $U_{1/2}$ directly. To this end, we

first note that $\Gamma' = \tau_z \otimes \Gamma$ is also a chiral symmetry with respect to $\tilde{U}_{1/2} = e^{-i\frac{\pi}{4}} U_{1/2} e^{i\frac{\pi}{4}}$, i.e., $U_{1/2}$ under the shift in the initial time from $t = 0$ to $t = 1/4$. Similar to its parent counterpart, such a chiral symmetry is responsible for protecting $E' = 0, \pm\pi$ quasienergy edge states in the square-root system. Next, we identify an additional symmetry $\Gamma'_{1/2} = \tau_z \otimes \mathcal{I}$ ($\mathcal{I}$ being the identity operator), which acts only within the enlarged degree of freedom and is thus referred to as the "subchiral" symmetry [79]. Such a symmetry operates as $\Gamma'_{1/2} U_{1/2} \Gamma'^{\dagger}_{1/2} = -U_{1/2}$. Consequently, if $|\psi'\rangle$ is a quasienergy $E$ eigenstate of $U_{1/2}$, then $\Gamma'_{1/2}|\psi'\rangle$ is a quasienergy $E \pm \pi$ eigenstate of $U_{1/2}$. Indeed,

$$U_{1/2} \Gamma'_{1/2} |\psi'\rangle = -\Gamma'_{1/2} U_{1/2} \Gamma'^{\dagger}_{1/2} \Gamma'_{1/2} |\psi'\rangle = e^{-i(E\pm\pi)} \Gamma'_{1/2} |\psi'\rangle .$$

In this case, a quasienergy which satisfies $E \pm \pi = -E$, i.e., $E = \pm\pi/2$, is necessarily twofold degenerate due to the product $\Gamma' \Gamma'_{1/2}$. The associated quasienergy eigenstates could further be chosen to be simultaneous $\pm 1$ eigenstates of $\Gamma' \Gamma'_{1/2}$. This is automatically the case for the quasienergy $\pm\pi/2$ edge/corner states. In particular, since $\pm 1$ eigenstates of $\Gamma' \Gamma'_{1/2}$ correspond to states localized at two opposite edges/corners, the discreteness of $\Gamma' \Gamma'_{1/2}$ eigenstates pins such edge/corner states at quasienergy $\pm\pi/2$ in the presence of symmetry-preserving perturbations. This completes the symmetry protection analysis of quasienergy $\pm\pi/2$ edge/corner states in the square-root system.

The above argument can be easily extended to conclude that, for any $q$th-root version of the system, $U_{1/q}^q$ also possesses the CS with respect to $\Gamma'' = \eta_z \otimes \Gamma$. A generalized "subchiral" symmetry can further be identified as $\Gamma''_{1/q} = \eta_z \otimes \mathcal{I}$, which operates as $\Gamma''_{1/q} U_{1/q} \Gamma''^{\dagger}_{1/q} = \omega^{\dagger} U_{1/q}$ and thus forces the quasienergies of $U_{1/q}$ to form a cluster of $E - 2\pi j/q$ with $j = 0, 1, \cdots, q-1$. In the case of cubic-root Floquet systems, which are explicitly studied below, both symmetries lead to the protection of degenerate edge/corner modes at $E'' = 0, \pm\pi/3, \pm 2\pi/3, \pm\pi$. Following the same routine, we can deduce that if the parent model $U$ possesses the CS $\Gamma$, the existence of its edge/corner modes at the quasienergies $E = 0, \pi$ guarantees the presence of degenerate edge/corner states at the quasienergies $(0, 1, ...2^n)\pi/2^n$ and $(0, 1, ..., 3^n)\pi/3^n$ of the systems described by $U_{1/2^n}$ and $U_{1/3^n}$, respectively. Notably, the boundary modes appearing at the fractional quasienergies $p\pi/q$ with $p < q$ and $(p, q)$ being co-prime integers are, to the best of our knowledge, not identified by previous studies on the symmetry classification and bulk-boundary correspondence of Floquet systems. They are thus a unique product of our $q$th-root procedure operated on Floquet operators. In Sec. 4, we will apply our theory to explicitly construct square/cubic-root first- and second-order non-Hermitian FTIs based on the parent models defined in the following section.

## 3 Parent models

In this section, we introduce two non-Hermitian Floquet topological insulator (FTI) models that will be taking square- and cubic-roots. Detailed investigations of these parent models can be found in Refs. [62] and [66]. All system parameters below are assumed to be properly scaled and set in dimensionless units.

The first model of our consideration describes a non-Hermitian FTI with rich topological phase diagrams and arbitrarily many degenerate edge modes in the presence of Floquet NHSE [66]. Its time-dependent Hamiltonian is $H(t) = H_1$ for $t \in [\ell + 1/2, \ell + 1)$ and $H(t) = H_2$ for $t \in [\ell, \ell + 1/2)$, where $t$ denotes time, $\ell \in \mathbb{Z}$, and

$$H_1 = \sum_n J_2(i|n+1\rangle\langle n| + \text{H.c.}) \otimes \sigma_y + \sum_n i\lambda(|n\rangle\langle n+1| + \text{H.c.}) \otimes \sigma_y, \tag{19}$$

$$H_2 = \sum_n [2\mu|n\rangle\langle n| + J_1(|n\rangle\langle n+1| + \text{H.c.})] \otimes \sigma_x + \sum_n i\lambda(i|n+1\rangle\langle n| + \text{H.c.}) \otimes \sigma_x. \tag{20}$$

Here $n \in \mathbb{Z}$ is the unit cell index. $\sigma_{x,y,z}$ are Pauli matrices acting on the two sublattices in each unit cell. $J_{1,2}$ and $i\lambda$ describe symmetric and asymmetric parts of intercell hopping amplitudes. $\mu$ is the intracell coupling strength. The Floquet operator $U = e^{-i\frac{1}{2}H_1}e^{-i\frac{1}{2}H_2}$ that governs the evolution of the system over a driving period (e.g., from $t = 0$ to 1) is nonunitary once $\lambda \neq 0$. This yields a model that could possess non-Hermitian FTI phases, which are characterized by integer or half-integer quantized topological invariants under the periodic boundary conditions (PBC) [66]. Under the open boundary conditions (OBC), the CS of the model $\Gamma = \mathbb{I}_N \otimes \sigma_z$ ($N$ is the number of unit cells and $\mathbb{I}_N$ is an $N \times N$ identity) allows multiple edge modes to appear in pairs at the quasienergies zero and $\pi$, whose numbers can be determined by the OBC bulk winding numbers $\nu_0$ and $\nu_\pi$ (see Sec. A for their definitions). These edge modes are further found to coexist with sufficient amounts of bulk states localized around both edges of the system due to the NHSE [66].

The second model that we will employ describes a non-Hermitian Floquet second-order topological insulator (FSOTI), which could possess multiple quartets of corner-localized states at real quasienergies zero and $\pi$ [62]. The Hamiltonian of the model takes the form of $H(t) = \mathcal{H}_1$ for $t \in [\ell + 1/2, \ell + 1)$ and $H(t) = \mathcal{H}_2$ for $t \in [\ell, \ell + 1/2)$ with $\ell \in \mathbb{Z}$. Here

$$\mathcal{H}_{1(2)} = \mathcal{H}_x \otimes \mathbb{I}_y + \mathbb{I}_x \otimes \mathcal{H}_{y1(y2)}, \tag{21}$$

$$\mathcal{H}_x = \Delta \sum_{m,n}(|m,n\rangle\langle m+1,n| \otimes \sigma_- + \text{H.c.}), \tag{22}$$

$$\mathcal{H}_{y1} = \sum_{m,n}(iJ_2|m,n+1\rangle\langle m,n| + \text{H.c.} + 2\mu|m,n\rangle\langle m,n|) \otimes \sigma_z, \tag{23}$$

$$\mathcal{H}_{y2} = J_1 \sum_{m,n}(|m,n\rangle\langle m,n+1| + \text{H.c.}) \otimes \sigma_x. \tag{24}$$

The $\mathbb{I}_x$ and $\mathbb{I}_y$ are identity matrices for the basis along $x$ and $y$ directions of the lattice. $\sigma_{x,y,z}$ are Pauli matrices and $\sigma_- = (\sigma_x - i\sigma_y)/2$. $m,n \in \mathbb{Z}$ are unit cell indices along the two spatial dimensions. $\Delta$ and $J_{1,2}$ describe hopping amplitudes between nearest neighbor cells along the $x$ and $y$ directions. $\mu$ characterizes the strength of an onsite potential bias. Gain and loss are introduced to make the system non-Hermitian by setting $\mu = u + iv$, with $u,v \in \mathbb{R}$ and $v \neq 0$. The Floquet operator of the system takes the form $\mathcal{U} = e^{-i\frac{1}{2}\mathcal{H}_1}e^{-i\frac{1}{2}\mathcal{H}_2}$, whose spectrum under the OBC features fourfold degenerate topological corner modes at zero and $\pi$ quasienergies. The numbers of these corner modes $n_0$ and $n_\pi$ are related to a pair of bulk topological winding numbers $\nu_0$ and $\nu_\pi$ of $\mathcal{U}$ (see Sec. B for their definitions) through a bulk-corner correspondence relation $(n_0, n_\pi) = 4(|\nu_0|, |\nu_\pi|)$ [62]. The fourfold degeneracy of Floquet corner modes at the quasienergies $E = 0, \pi$ is protected by the CS $\Gamma = \sigma_z \otimes \sigma_y$ of the two-dimensional system described by $\mathcal{U}$ under the PBC [62].

Applying the procedure of Sec. 2, we will obtain the square and cubic roots of the two Floquet models introduced here, and unveil the intriguing topological features of the resulting systems in the following section. As will be demonstrated, our $q$th-root procedure endows the non-Hermitian Floquet phases in the above two parent models with even richer topological properties.

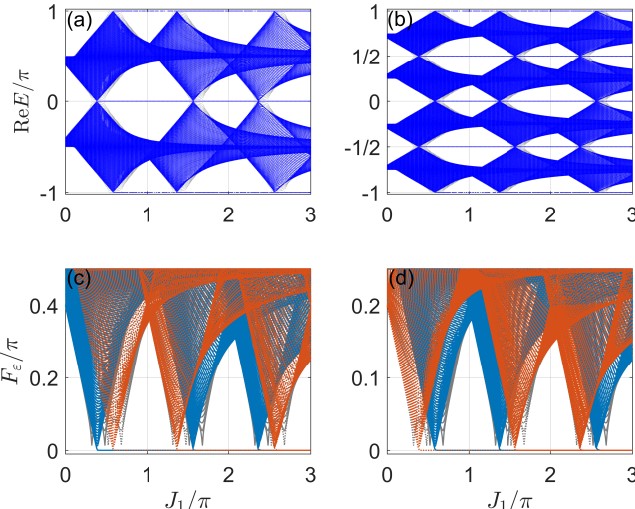

Figure 2: Floquet spectrum $E$ and gap function $F_\varepsilon$ of $U_{1/2}$ [Eq. (25)] and $U$ versus $J_1$ under OBC and PBC. Other system parameters are set as $(J_2, \mu, \lambda) = (0.5\pi, 0.4\pi, 0.25)$ and the length of lattice is $L = 400$. (a) and (b) show the values of the real part of $E$ for the parent and square-root models described by $U$ and $U_{1/2}$ under the OBC (PBC) in blue (grey) dots, respectively. The blue solid and red dotted lines denote the gap functions $F_0$ and $F_\pi$ of $U$ in (c), and the gap functions $F_0$ ($= F_\pi$) and $F_{\pi/2}$ of $U_{1/2}$ in (d) under the OBC. Grey solid and dotted lines denote the same gap functions under PBC.

## 4 Results

In Sec. 4.1, we present square- and cubic-root non-Hermitian FTIs generated by the first model in Sec. 3, which will be shown to possess multiple and tunable numbers of degenerate edge modes with the quasienergies $\pi/2$, $\pi/3$ and $2\pi/3$ that could survive under the NHSE. In Sec. 4.2, we discuss square- and cubic-root non-Hermitian FSOTIs yielded by the second model in Sec. 3, which hold non-Hermiticity induced quartets of topological corner modes at the $\pi/2$, $\pi/3$ and $2\pi/3$ quasienergies.

### 4.1 Square/Cubic-root non-Hermitian FTIs

We now apply the procedure in Sec. 2 to find the square- and cubic-roots of the first model in Sec. 3. In the lattice representation, the square-root Floquet system is obtained by identifying $U_1 = e^{-iH_1/2}$ and $U_2 = e^{-iH_2/2}$ in Eq. (4), where the $H_1$ and $H_2$ are given by Eqs. (19) and (20), respectively. The Floquet operator $U_{1/2}$ is then derived following Eq. (1), i.e.,

$$U_{1/2} = \begin{pmatrix} 0 & -e^{-iH_1/2} \\ e^{-iH_2/2} & 0 \end{pmatrix}. \tag{25}$$

To obtain the cubic-root model, we may identify $\tilde{H}_1 = \tilde{H}_3 = 3H_1/4$ and $\tilde{H}_2 = 3H_2/2$ in Eq. (17), where $H_1$ and $H_2$ are defined by Eqs. (19) and (20), respectively. This then leads to the Floquet operator

$$U_{1/3} = \begin{pmatrix} 0 & e^{-iH_1/4} & 0 \\ 0 & 0 & e^{-iH_2/2} \\ e^{-iH_1/4} & 0 & 0 \end{pmatrix}. \tag{26}$$

Solving the eigenvalue equations $U_{1/2(1/3)}|\psi\rangle = e^{-iE}|\psi\rangle$ under the OBC, with $E$ being the quasienergy, provides us with all bulk and edge states of the square- (cubic-) root Floquet

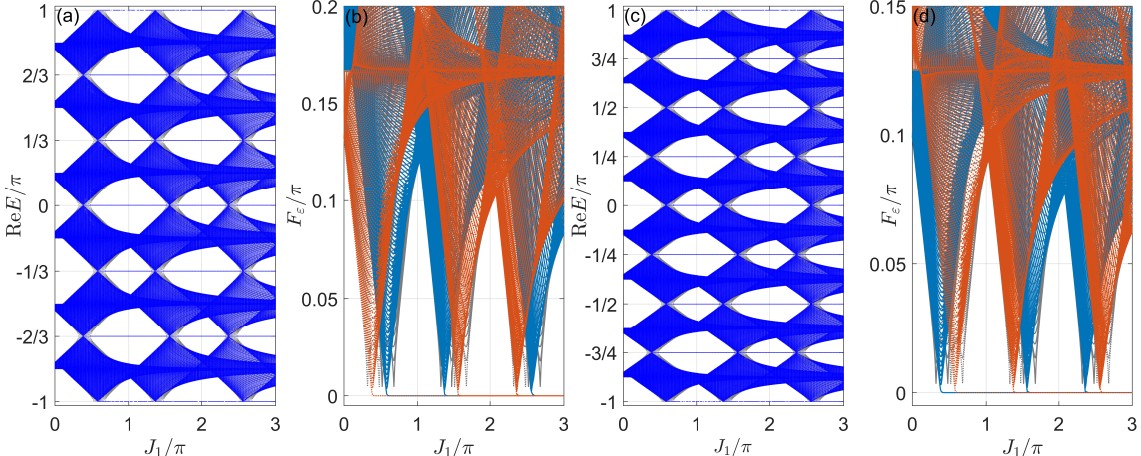

Figure 3: Floquet spectrum $E$ and gap function $F_\varepsilon$ of $U_{1/3}$ [Eq. (26)] and $U_{1/4}$ [obtained following Eqs. (25), (6) and (7)] versus $J_1$ under both PBC and OBC. Other system parameters and the lattice size are the same as those used in Fig. 2. (a) and (c) show the real parts of $E$ for the cubic- and fourth-root models described by $U_{1/3}$ and $U_{1/4}$, respectively, under the OBC (blue dots) and PBC (grey dots in the background). The blue solid and red dotted lines denote the gap functions $F_{\pi/3}$ ($= F_\pi$) and $F_{2\pi/3}$ ($= F_0$) of $U_{1/3}$ in (b), and the gap functions $F_{\pi/4}$ ($= F_{3\pi/4}$) and $F_{\pi/2}$ ($= F_0 = F_\pi$) of $U_{1/4}$ in (d) under the OBC. Corresponding gap functions under the PBC are given by the grey solid and dotted lines in (b) and (d).

system.

As an important note, if there is a pair of degenerate edge modes with zero-quasienergy in the parent model $U = e^{-iH_1/2}e^{-iH_2/2}$, their quasienergies will be shifted to $\pi$ in $U_{1/2}^2$ according to Eq. (5), yielding edge modes at quasienergies $E = \pm\pi/2$ in the system described by $U_{1/2}$. On the other hand, if a pair of degenerate edge modes appears at $E = \pi$ in the parent model, their quasienergies will become 0 (mod $2\pi$) in $U_{1/2}^2$, leading to edge states at $E = 0, \pm\pi$ in the square-root model. Following the same routine, we deduce that the edge modes at zero ($\pi$) quasienergy in $U = e^{-iH_2/4}e^{-iH_1/2}e^{-iH_2/4}$ generate edge states with $E = 0, \pm2\pi/3$ ($E = \pm\pi/3, \pm\pi$) in the system described by $U_{1/3}$ after taking the cubic root. Now if we could relate the numbers of zero and $\pi$ edge modes in the parent system $U$ to its topological invariants, these invariants should also predict the numbers of zero, $\pi/2$, $\pi/3$, $2\pi/3$ and $\pi$ modes in the square- and cubic-root systems if the symmetry that protects their quantization is not broken during the process of taking roots.

To showcase the fractional-quasienergy edge modes in the spectrum in a more transparent manner, we introduce the gap function $F_\varepsilon$ with respect to a quasienergy $\varepsilon$, which is defined as

$$F_\varepsilon = \sqrt{(\mathrm{Re}E - \varepsilon)^2 + (\mathrm{Im}E)^2}. \tag{27}$$

Note that the $E$ in Eq. (27) is the collection of all quasienergies obtained by diagonalizing the Floquet operator of the system under consideration. It is clear that once there is an edge state with real quasienergy $\varepsilon$ that resides in a gap on the complex plane, we would have $F_\varepsilon = 0$ for that state and $F_\varepsilon > 0$ for all other bulk states. To locate the expected edge states of $U_{1/2}$ and $U_{1/3}$, we choose $\varepsilon = 0, \pi/2, \pi$ and $\varepsilon = 0, \pi/3, 2\pi/3, \pi$ for them, respectively, in the following numerical calculations.

In Fig. 2, we present the quasienergy (Floquet) spectrum and gap functions of the first model in Sec. 3 and its square-root descendant under both the PBC and OBC. The quasienergies and gap functions of the parent model in Figs. 2(a) and 2(c) are reproduced

from Ref. [66]. A clear distinction between the spectrum under PBC (gray dots in the background) and OBC (blue dots) can be observed especially around the phase transition points, implying the presence of NHSE in the system. To retrieve the bulk-edge correspondence, a pair of open-boundary winding numbers $(\nu_0, \nu_\pi)$ is introduced in Ref. [66] and reviewed in Sec. A, which correctly counts the number of zero- and $\pi$-quasienergy edge modes $n_0$ and $n_\pi$ in the parent model through the relation $(n_0, n_\pi) = 2(|\nu_0|, |\nu_\pi|)$. Here $n_E$ denotes the number of edge states at the quasienergy $E$. According to our square-root procedure, the edge modes at the quasienergies $E = 0, \pm\pi$ ($E = \pm\pi/2$) are generated by taking the square-root of the $\pi$ (zero) Floquet edge modes. Therefore, we arrive at the following bulk-edge correspondence for the square-root FTIs described by $U_{1/2}$, i.e.,

$$n_{\pi/2} = 2|\nu_0|, \qquad n_0 = n_\pi = 2|\nu_\pi|, \tag{28}$$

where $n_{\pi/2}$ means the number of degenerate edge states at $E = \pm\pi/2$. These relations are readily confirmed by comparing the spectrum and gap functions presented in Figs. 2(b,d) and Figs. 2(a,c). Notably, with the increase of hopping amplitude $J_1$, we observe a series of gap closing and topological phase transitions in the square-root model. After each transition, the number of edge modes $n_0$, $n_\pi$ or $n_{\pi/2}$ is found to be increased by 2 even in the presence of NHSE. Specifically, we find $(n_{\pi/2}, n_0, n_\pi) = (0,0,0), (2,0,0), (2,2,2), (2,4,4), (4,4,4), (6,4,4), (6,6,6)$ with the increase of $J_1$ in Fig. 2(d), meanwhile the winding numbers are $(\nu_0, \nu_\pi) = (0,0), (1,0), (1,-1), (1,-2), (2,-2), (3,-2), (3,-3)$ according to the calculation reported in Ref. [66]. This process could continue with the further increase of $J_1$. We can thus in principle obtain arbitrarily many topological edge modes at fractional quasienergies $E = \pm\pi/2$ in our square-root non-Hermitian Floquet system. This highlights the universal advantage of Floquet engineering in generating unique nonequilibrium states with strong topological signatures.

In Figs. 3(a) and 3(b), we further show the Floquet spectrum and gap function of the cubic-root non-Hermitian FTI. In addition to edge states at the quasienergies $E = 0, \pm\pi$, we also observe degenerate edge modes at fractional quasienergies $E = \pm\pi/3, \pm 2\pi/3$. Recall that Eq. (26) cubes to a block diagonal matrix consisting of multiple copies of Floquet operator of the parent model $U$. Therefore, the edge states at quasienergies $(0, \pm 2\pi/3)$ $[(\pm\pi/3, \pm\pi)]$ are indeed descendants of the zero $[\pi]$ edge modes in the parent model, whose numbers are counted by $\nu_0$ $[\nu_\pi]$ [66]. We then obtain the bulk-edge correspondence for cubic-root non-Hermitian FTIs as

$$n_0 = n_{2\pi/3} = 2|\nu_0|, \qquad n_{\pi/3} = n_\pi = 2|\nu_\pi|. \tag{29}$$

With the increase of $J_1$, the cubic-root system could also undergo a series of topological phase transitions, with each of them being accompanied by the increase of either $n_0$ and $n_{2\pi/3}$ or $n_{\pi/3}$ and $n_\pi$ by two. We can thus obtain arbitrarily many $\pi/3$ and $2\pi/3$ degenerate edge modes by tuning the single driving parameter $J_1$ even in the presence of NHSE. Since it has been demonstrated that Floquet edge states could be utilized to construct boundary discrete time crystals (DTCs) [22, 80], we expect the emergence of unique non-Hermitian Floquet boundary DTCs through the superposition of $(\pm\pi/3, \pm 2\pi/3)$ edge modes and other edge states in the cubic-root FTIs.

For completeness, we present the spectrum and gap function in Figs. 3(c) and 3(d) for the fourth-root non-Hermitian FTI, which is constructed by applying the procedure in Eqs. (6) and (7) to the first model in Sec. 3. The resulting system holds Floquet edge states at $E = \pm\ell\pi/4$ with $\ell = 0, ..., 4$. Similar to our analysis of $U_{1/2}$ and $U_{1/3}$, the number of these edge modes are related to the bulk topological invariants $(\nu_0, \nu_\pi)$ of the parent Floquet system via

$$n_{\pi/4} = n_{3\pi/4} = 2|\nu_0|, \qquad n_0 = n_{\pi/2} = n_\pi = 2|\nu_\pi|. \tag{30}$$

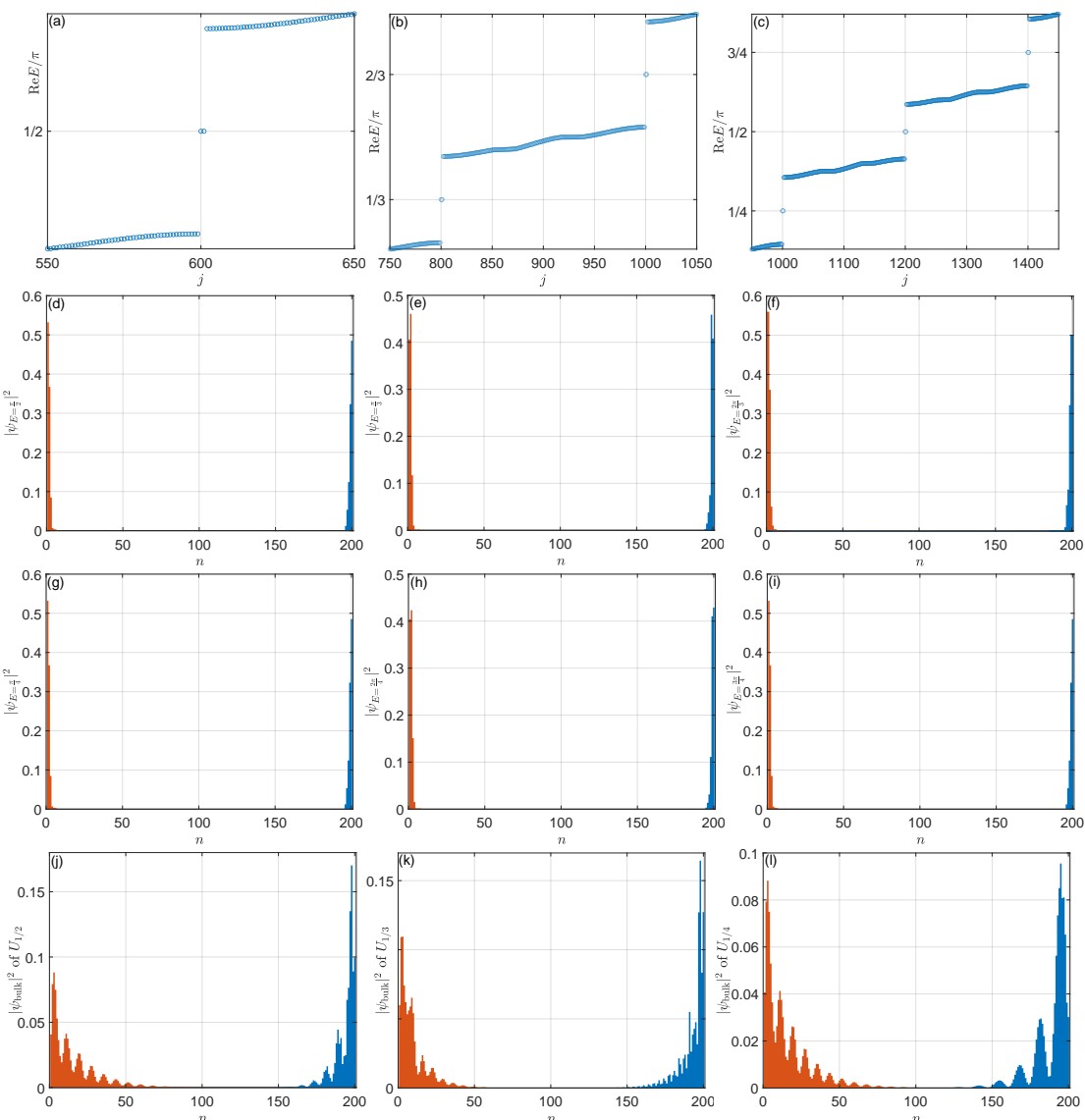

Figure 4: The real part of Floquet spectrum, fractional-quasienergy edge modes and bulk skin modes of the $q$th-root non-Hermitian FTIs for $q = 2, 3, 4$. $j$ and $n$ are state and unit cell indices. System parameters are $(J_1, J_2, \mu, \lambda) = (\pi, 0.5\pi, 0.4\pi, 0.25)$. The length of lattice is $L = 400$. (a), (b) and (c) show the Floquet spectrum of $U_{1/2}$, $U_{1/3}$ and $U_{1/4}$, zoomed in around $E = \pi/2$, $E = (\pi/3, 2\pi/3)$ and $E = (\pi/4, 2\pi/4, 3\pi/4)$, respectively. (d) shows the degenerate edge modes of $U_{1/2}$ with $E = \pi/2$. (e) and (f) show the degenerate edge modes of $U_{1/3}$ with $E = \pi/3$ and $2\pi/3$. (g), (h) and (i) show the degenerate edge modes of $U_{1/4}$ with $E = \pi/4$, $2\pi/4$ and $3\pi/4$. (j), (k) and (l) show three pairs of typical bulk states for $U_{1/2}$, $U_{1/3}$ and $U_{1/4}$, respectively, which are piled up around the boundaries and thus represent non-Hermitian Floquet skin modes.

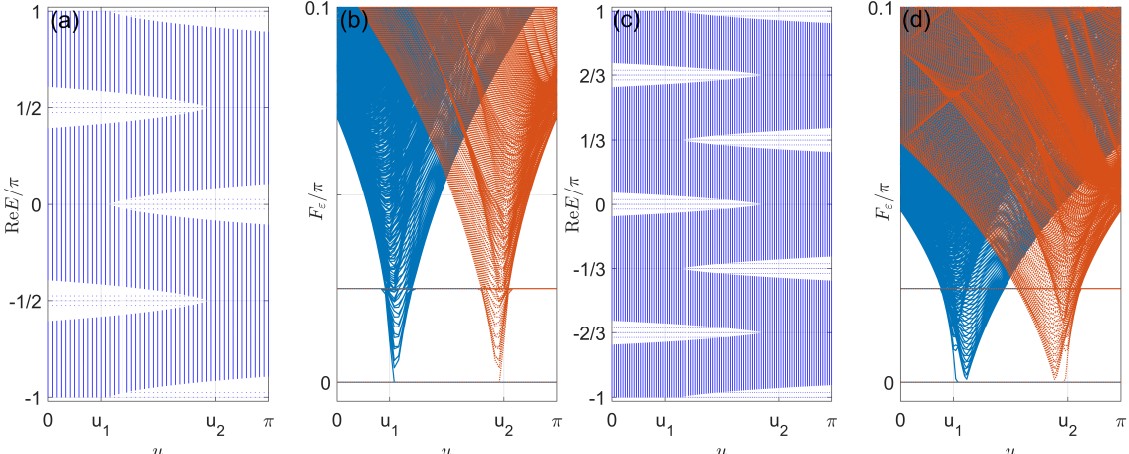

Figure 5: Floquet spectrum $E$ and gap function $F_\varepsilon$ of $\mathcal{U}_{1/2}$ [Eq. (31)] and $\mathcal{U}_{1/3}$ [Eq. (32)] versus $u$. Other system parameters are set as $(J_1, J_2, \Delta, v) = (0.5\pi, 5\pi, 0.05\pi, 0.5)$. The lattice size is $L = 2000$ along each spatial direction. (a) and (c) show the values of the real part of $E$ for the square- and cubic-root models described by $\mathcal{U}_{1/2}$ and $\mathcal{U}_{1/3}$, respectively. The blue solid and red dotted lines denote the gap functions $F_0$ ($= F_\pi$) and $F_{\pi/2}$ of $\mathcal{U}_{1/2}$ in (b), and the gap functions $F_{\pi/3}$ ($= F_\pi$) and $F_{2\pi/3}$ ($= F_0$) of $\mathcal{U}_{1/3}$ in (d). $u_1$ and $u_2$ denote critical values of $u$ where the spectral gap closes and the number of corner modes changes across the topological phase transition, which are obtained from Eq. (33).

More precisely, we find the values of $(n_E, n_{E'})$ for the rooted model to change in the sequence $(0,0), (2,0), (2,2), (2,4), (4,4), (6,4), (6,6)$ for $E = 0, 2\pi/3$ ($E = \pi/4, 3\pi/4$) and $E' = \pi/3, \pi$ ($E' = 0, \pi/2, \pi$) in Fig. 3(b) (Fig. 3(d)) with the increase of $J_1$, while the winding numbers of the parent model change as $(v_0, v_\pi) = (0,0), (1,0), (1,-1), (1,-2), (2,-2), (3,-2), (3,-3)$ during the process [66], confirming the relations in Eqs. (29) and (30). In Fig. 4, we present examples of degenerate edge modes at fractional quasienergies and bulk skin modes that co-exist with these topological edge states in the systems described by $U_{1/q}$ for $q = 2, 3, 4$. The numbers of edge modes found there coincide with our theoretical predictions. All these results reveal the power of our strategy in constructing $q$th-root FTIs for any $q \in \mathbb{Z}^+$. In the following subsection, we will further demonstrate the applicability of the same routine in the construction of $q$th-root FSOTIs.

## 4.2 Square/Cubic-root non-Hermitian Floquet second order topological insulators

We next generate square- and cubic-root non-Hermitian Floquet second-order topological insulators (FSOTIs) by applying our theory in Sec. 2 to the second model in Sec. 3. To find the square-root model, we identify $U_1 = e^{-i\mathcal{H}_1/2}$ and $U_2 = e^{-i\mathcal{H}_2/2}$ in Eq. (4), where the $\mathcal{H}_1$ and $\mathcal{H}_2$ are given by Eq. (21) in Sec. 3. The resulting square-root Floquet operator reads

$$\mathcal{U}_{1/2} = \begin{pmatrix} 0 & -e^{-i\mathcal{H}_1/2} \\ e^{-i\mathcal{H}_2/2} & 0 \end{pmatrix}. \tag{31}$$

Similarly, to obtain the cubic-root model, we identify $\tilde{H}_1 = \tilde{H}_3 = 3\mathcal{H}_1/4$ and $\tilde{H}_2 = 3\mathcal{H}_2/2$, where the $\mathcal{H}_1$ and $\mathcal{H}_2$ are given by Eq. (21). This leads to the cubic-root Floquet

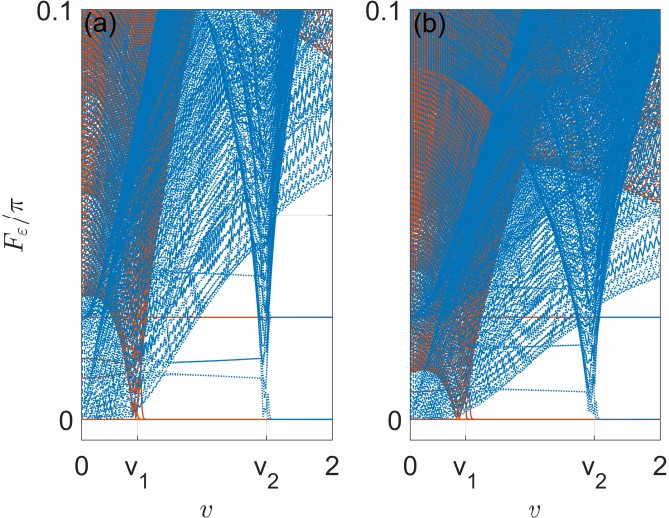

Figure 6: Gap function $F_\varepsilon$ of $\mathcal{U}_{1/2}$ [Eq. (31)] and $\mathcal{U}_{1/3}$ [Eq. (32)] versus $v$. Other system parameters are $(J_1, J_2, \Delta, u) = (2.5\pi, 3\pi, 0.05\pi, 0)$ and the length of lattice is $L = 2000$ in each spatial dimension. The blue dotted and red solid lines denote the gap functions $F_0 (= F_\pi)$ and $F_{\pi/2}$ of $\mathcal{U}_{1/2}$ in (a), and the gap functions $F_{\pi/3} (= F_\pi)$ and $F_{2\pi/3} (= F_0)$ of $\mathcal{U}_{1/3}$ in (b). $v_1$ and $v_2$ are imaginary parts of $\mu = u + iv$ where the spectral gap closes and the number of corner modes change through topological phase transitions, obtained by solving Eq. (33).

operator

$$
\mathcal{U}_{1/3} = \begin{pmatrix} 0 & e^{-i\mathcal{H}_1/4} & 0 \\ 0 & 0 & e^{-i\mathcal{H}_2/2} \\ e^{-i\mathcal{H}_1/4} & 0 & 0 \end{pmatrix}.
\tag{32}
$$

Recall that the parent Floquet system $\mathcal{U} = e^{-i\frac{1}{2}\mathcal{H}_1}e^{-i\frac{1}{2}\mathcal{H}_2}$ possesses multiple and non-Hermiticity induced fourfold degenerate corner modes at zero and $\pi$ quasienergies, which are obtained by solving the eigenvalue equation $\mathcal{U}|\psi\rangle = e^{-iE}|\psi\rangle$. The numbers of these corner modes are related to a pair of topological invariants introduced in Ref. [62] (see also Sec. B). Since the process of taking the square (cubic) root of $\mathcal{U}$ does not break the protecting CS of these corner modes, we expect the topological invariants of $\mathcal{U}$ to be able to predict the numbers of corner modes at the $(0, \pi/2, \pi)$ $[(0, \pi/3, 2\pi/3, \pi)]$ quasienergies of $\mathcal{U}_{1/2}$ $[\mathcal{U}_{1/3}]$.

The spectra of $\mathcal{U}_{1/2}$ and $\mathcal{U}_{1/3}$ are presented in Fig. 5. In Figs. 5(a) and 5(c), we observe states at the fractional-quasienergies $E = \pm\pi/2$ and $E = \pm\pi/3, \pm 2\pi/3$ for the square-root and cubic-root systems respectively, whose spatial profiles are found to be localized around the four corners of the lattice. The numbers of these corner modes can further be controlled by changing the real part of onsite potential $u$, as can be seen clearly in Figs. 5(b) and 5(d). The critical values $(u_1, u_2)$ where the number of corner modes change correspond to topological phase transition points, which are determined by the gapless condition of the parent model [62], i.e.,

$$
\cos\left[J_1\sqrt{1 - (n\pi - u)^2/J_2^2}\right]\cosh v = \pm 1.
\tag{33}
$$

Moreover, since the $\pi/2$ (zero and $\pi$) modes of $\mathcal{U}_{1/2}$ are inherited from the zero $(\pi)$ modes of the parent model $\mathcal{U}$, their numbers are determined by the winding numbers $(v_0, v_\pi)$ of the second model in Sec. 3 (see also Sec. B) through the bulk-corner correspondence relations

$$
n_{\pi/2} = 4|v_0|, \qquad n_0 = n_\pi = 4|v_\pi|.
\tag{34}
$$

Figure 7: Gap functions and probability distributions of corner modes for $\mathcal{U}_{1/2}$ in (a), (c)–(e) and for $\mathcal{U}_{1/3}$ in (b), (f)–(j). $j$ and $n_{x,y}$ are state and unit cell indices. System parameters are $(J_1, J_2, \Delta, \mu) = (2.5\pi, 3\pi, 0.05\pi, 2i)$. The lattice size is $L_x = L_y = 3000$. (a) and (b) show the gap function $F_\varepsilon$, which is zoomed in around $F_\varepsilon = 0$ for $\varepsilon = \pi/2$, $\pi/3$ and $2\pi/3$. (c)–(e) show the twelve corner modes of $\mathcal{U}_{1/2}$ at the quasienergy $\pi/2$ (with $F_{\pi/2} = 0$). (f), (g) show the eight corner modes of $\mathcal{U}_{1/3}$ at the quasienergy $\pi/3$ (with $F_{\pi/3} = 0$). (h)–(j) show the twelve corner modes of $\mathcal{U}_{1/3}$ at the quasienergy $2\pi/3$ (with $F_{2\pi/3} = 0$).

Similarly, as the zero and $2\pi/3$ ($\pi/3$ and $\pi$) corner modes of $\mathcal{U}_{1/3}$ are inherited from the zero ($\pi$) corner modes of $\mathcal{U}$, we have the following relations to determine their numbers from the bulk invariants of the parent model, i.e.,

$$n_0 = n_{2\pi/3} = 4|\nu_0|, \qquad n_{\pi/3} = n_\pi = 4|\nu_\pi|. \tag{35}$$

In the regions $u \in (0, u_1), (u_1, u_2), (u_2, \pi)$, we find winding numbers $(\nu_0, \nu_\pi) = (5, 4), (5, 5),$ $(4, 5)$, and the number of corner modes $(n_{\pi/2}, n_0, n_\pi) = (20, 16, 16), (20, 20, 20), (16, 20, 20)$ $[(n_0, n_{2\pi/3}, n_{\pi/3}, n_\pi) = (20, 20, 16, 16), (20, 20, 20, 20), (16, 16, 20, 20)]$ for $\mathcal{U}_{1/2}$ [$\mathcal{U}_{1/3}$], which confirm the Eqs. (34) and (35).

A very intriguing feature of the square- and cubic-root FSOTIs studied here is that the non-Hermitian effect could induce more topological corner modes than in the Hermitian limit. To demonstrate this, we investigate the gap functions of $\mathcal{U}_{1/2}$ and $\mathcal{U}_{1/3}$ versus the gain and loss strength $v$ in Figs. 6(a) and 6(b). In both figures, $v_1$ and $v_2$ denote critical values of $v$ where the system undergoes topological phase transitions. Their specific values are determined by the gapless condition of the parent model in Eq. (33). With the increase of $v$ from 0 to 2, we find that in the three regions $v \in (0, v_1), (v_1, v_2), (v_2, 2)$, the winding numbers of the parent model are $(\nu_0, \nu_\pi) = (1, 0), (3, 0), (3, -2)$ [62], whereas the number of corner modes are $(n_{\pi/2}, n_0, n_\pi) = (4, 0, 0), (12, 0, 0), (12, 8, 8)$ in Fig. 6(a) for the square-root model $\mathcal{U}_{1/2}$, and $(n_0, n_{2\pi/3}, n_{\pi/3}, n_\pi) = (4, 4, 0, 0), (12, 12, 0, 0), (12, 12, 8, 8)$ in Fig. 6(b) for the cubic-root system $\mathcal{U}_{1/3}$. These results are clearly consistent with the bulk-corner correspondence relations for square- and cubic-root FSOTIs as stated in Eqs. (34) and (35).

Note in passing that the horizontal lines appearing at $F_\varepsilon \neq 0$ in Figs. 5(b), 5(d) and Fig. 6 are related to eigenmodes formed by coupling edge states along the $y$-direction and bulk states along the $x$-direction of the lattice. As the 1D chains along $y$ possess a chiral symmetry, the coupling of their degenerate edge modes with the bulk states along $x$ can yield degenerate states in 2D that are protected by the chiral symmetry of a 1D subsystem. We may thus regard these edge states as weak edge states caused by weak topology. In the 2D system, their number is sensitive to the system size along $x$ and their quasienergies are sensitive to the choice of system parameters. Comparatively, the numbers and quasienergies of corner modes are solely protected by the chiral symmetry and topological invariants of the 2D system, making them robust to the changes of system size and parameters before encountering a phase transition.

Finally, in Fig. 7, we present the gap functions and spatial profiles of Floquet corner modes at the quasienergies $\pi/2$, $\pi/3$ and $2\pi/3$. Their numbers are found to precisely coincide with the bulk-corner correspondence relations in Eqs. (34) and (35). The non-Hermiticity enriched higher-order topology in rooted Floquet systems may also assist us to engineer unique DTCs and quantum computing schemes with the multiple quartets of corner modes at different fractional-quasienergies that are robust to the perturbation of environment.

## 5 Conclusion

In summary, we proposed a systematic approach to construct the $q$th-root of any periodically driven system and presented $2^n$th- and $3^n$th-root Floquet topological insulators as explicit examples. The latter are shown to exhibit degenerate edge/corner modes at fractional quasienergies $\frac{\pi}{2^n}(0, 1, ..., 2^n)$ and $\frac{\pi}{3^n}(0, 1, ..., 3^n)$, whose topological nature is inherited from their $2^n$th- and $3^n$th-power parent systems. Square- and cubic-root non-Hermitian Floquet topological insulators with multiple and tunable topological edge/corner states at quasienergies $\pi/2$, $\pi/3$ and $2\pi/3$ were investigated in details. Further connections were made between the number of these edge/corner modes and the bulk topological invariants of parent systems, yielding the bulk-edge/corner correspondence in two classes of rooted Floquet topological insulators.

Intriguingly, non-Hermitian effects are found to induce more corner modes with fractional-quasienergies and generate multiple edge states coexisting with the non-Hermitian skin effect in rooted systems. Our discoveries thus uncover a unique class of topological phases that originates from the cooperation among driving, non-Hermiticity and the process of taking the nontrivial roots of Floquet systems.

From the experimental perspective, the obtained systems from our $q$th-rooting procedure are expected to be implementable with the same setups employed for realizing their parent models. To this end, the additional degrees of freedom required in our $q$th-rooting procedure can be principally implemented by coupling multiple copies of the parent system. In the context of the non-Hermitian Floquet first and second order topological insulators explored in this paper, their corresponding square- and cubic-root systems can thus be realized in principle via setups like photonic quantum walks [78, 81–84]. For example, the anisotropic hopping amplitude and non-Hermitian lattice potential can be implemented by introducing controlled optical losses with acousto-optical modulators in coupled optical fibre loops [78]. Moreover, the winding numbers used in characterizing their topology can in principle be experimentally probed via measuring mean chiral displacements [59, 61, 85] or time-averaged spin textures [60, 86], which can also be conducted in similar photonic setups [81–84].

In future work, it would be interesting to apply our scheme to realize $q$th-root chiral Floquet topological insulators and gapless topological phases in higher spatial dimensions. The application of our approach to systems with many-body interactions is also expected to be fruitful. In particular, it was recently shown that the interplay between interaction and periodic driving may promote $2\pi/2^n$ modes into $Z_{2^n}$ parafermions [87]. Other families of $2\pi/q$ modes obtained in this work thus open avenues for exploring different types of Floquet parafermions not covered in Ref. [87]. In particular, $Z_3$ parafermions, which are expected to arise in systems with $2\pi/3$ modes when subjected to appropriate interactions, form a main ingredient for constructing the powerful Fibonacci anyons [88] that enable topologically protected universal quantum computation.

## Acknowledgements

**Author contributions**   L.Z. and R.W.B. contributed equally to this work.

**Funding information**   L.Z. is supported by the National Natural Science Foundation of China (Grant No. 11905211), the Young Talents Project at Ocean University of China (Grant No. 861801013196), and the Applied Research Project of Postdoctoral Fellows in Qingdao (Grant No. 861905040009). R.W.B. is supported by the Australian Research Council Centre of Excellence for Engineered Quantum Systems (EQUS, CE170100009).

## A   Topological invariants of the non-Hermitian FTI

Here we briefly recap the open-boundary winding numbers (OBWNs) introduced in Ref. [66]. They will be used to establish the bulk-edge correspondence for the first parent model in Sec. 3 and its $q$th-root descendants in Sec. 4.1. We first consider the dynamics of the model in two symmetric time frames, where the Floquet operator $U = e^{-iH_1/2}e^{-iH_2/2}$ is transformed to $U_a = e^{-iH_2/4}e^{-iH_1/2}e^{-iH_2/4}$ and $U_b = e^{-iH_1/4}e^{-iH_2/2}e^{-iH_1/4}$. Next we define the $Q$-matrix in the time frame $\alpha$ ($= a, b$) as $Q_\alpha = \sum_j(|\psi_{\alpha j}^+\rangle\langle\tilde{\psi}_{\alpha j}^+| - |\psi_{\alpha j}^-\rangle\langle\tilde{\psi}_{\alpha j}^-|)$. The right and left Floquet eigenvectors $|\psi_{\alpha j}^\pm\rangle$ and $\langle\tilde{\psi}_{\alpha j}^\pm|$ satisfy the eigenvalue equations $U_\alpha|\psi_{\alpha j}^\pm\rangle = e^{-i(\pm E_j)}|\psi_{\alpha j}^\pm\rangle$ and

$\langle\tilde{\psi}_{\alpha j}^{\pm}|U_\alpha = \langle\tilde{\psi}_{\alpha j}^{\pm}|e^{-i(\pm E_j)}$. An OBWN for $U_\alpha$ is then defined as $\nu_\alpha = \text{Tr}_\text{B}(\Gamma Q_\alpha[Q_\alpha, X])/L_\text{B}$. Here $\Gamma$ is the chiral symmetry (CS) operator. $X$ is the unit cell position operator. For a system with $L$ lattice sites, we decompose it into a bulk region and two edge regions at the left and right. The trace $\text{Tr}_\text{B}$ is taken over the bulk region, which contains $L_\text{B}$ lattice sites. The length of each edge region is $L_E = (L - L_\text{B})/2$, which should be chosen properly in order to avoid the obstruction of non-Hermitian skin effect. Finally, we define two OBWNs for a 1D non-Hermitian FTI with CS as $\nu_0 = (\nu_a + \nu_b)/2$ and $\nu_\pi = (\nu_a - \nu_b)/2$. These winding numbers can only take integer values. They are further related to the numbers of Floquet edge modes at zero and $\pi$ quasienergies $n_0$ and $n_\pi$ through the relations $(n_0, n_\pi) = 2(|\nu_0|, |\nu_\pi|)$. Following our analysis in the main text, $(\nu_0, \nu_\pi)$ could also count the numbers of fractional-quasienergy edge modes in the $q$th-root descendants of the parent model $U$.

## B  Topological invariants of the non-Hermitian FSOTI

Here we summarize the definition of bulk winding numbers for the second parent model in Sec. 3 of main text. Following Ref. [62], we first transform the Floquet operator $\mathcal{U}$ into two symmetric time frames $a$ and $b$ by shifting the initial time of evolution from $t = 0$ to $t = 1/4$ and $t = 3/4$ respectively. The Floquet operators in these time frames take the forms $\mathcal{U}_a = e^{-i\mathcal{H}_2/4}e^{-i\mathcal{H}_1/2}e^{-i\mathcal{H}_2/4}$ and $\mathcal{U}_b = e^{-i\mathcal{H}_1/4}e^{-i\mathcal{H}_2/2}e^{-i\mathcal{H}_1/4}$. Performing Fourier transforms from position to momentum representations, we obtain $\mathcal{U}_\alpha = \sum_{k_x,k_y}|k_x, k_y\rangle\mathfrak{U}_\alpha(k_x, k_y)\langle k_x, k_y|$ with $\alpha = a, b$. In the tensor product form, we have $\mathfrak{U}_\alpha(k_x, k_y) = \mathcal{U}_0(k_x) \otimes \mathcal{U}_\alpha(k_y)$, where $\mathcal{U}_0(k_x) = e^{-i\mathcal{H}_x(k_x)}$,

$$\mathcal{U}_a(k_y) = e^{-i\mathcal{H}_{y2}(k_y)/4}e^{-i\mathcal{H}_{y1}(k_y)/2}e^{-i\mathcal{H}_{y2}(k_y)/4} , \tag{36}$$

$$\mathcal{U}_b(k_y) = e^{-i\mathcal{H}_{y1}(k_y)/4}e^{-i\mathcal{H}_{y2}(k_y)/2}e^{-i\mathcal{H}_{y1}(k_y)/4} . \tag{37}$$

The $\mathcal{H}_x(k_x)$, $\mathcal{H}_{y1}(k_y)$ and $\mathcal{H}_{y2}(k_y)$ are Fourier transforms of the Eqs. (22)–(24) in the main text. $\mathfrak{U}_\alpha(k_x, k_y)$ has the CS in the sense that $\Gamma\mathfrak{U}_\alpha(k_x, k_y)\Gamma = \mathfrak{U}_\alpha^{-1}(k_x, k_y)$ for $\alpha = a, b$, where $\Gamma = \sigma_z \otimes \sigma_y$. In our model, $\mathcal{U}_0$ simply describes the evolution operator of a Su-Schrieffer-Heeger model in its topological flat band limit, which possesses a winding number $w = 1$. Taking the Taylor expansion of $\mathcal{U}_\alpha(k_y)$ yields $\mathcal{U}_\alpha(k_y) = \cos(\mathcal{E}) - i(d_{\alpha x}\sigma_x + d_{\alpha z}\sigma_z)$, for which another winding number can be defined as $w_\alpha = \int_{-\pi}^{\pi}\frac{dk_y}{2\pi}\frac{d_{\alpha x}\partial_{k_y}d_{\alpha z} - d_{\alpha z}\partial_{k_y}d_{\alpha x}}{d_{\alpha x}^2 + d_{\alpha z}^2}$ for $\alpha = a, b$. Put together, we obtain a pair of winding numbers $(\nu_a, \nu_b) = w \times (w_a, w_b)$ for the Floquet operators $(\mathcal{U}_a, \mathcal{U}_b)$. Their combination results in the integer topological invariants $\nu_0 = (\nu_a + \nu_b)/2$ and $\nu_\pi = (\nu_a - \nu_b)/2$ of the two-dimensional parent system $\mathcal{U}$, which are related to the numbers of Floquet corner modes at zero and $\pi$ quasienergies $n_0$ and $n_\pi$ through the relations $(n_0, n_\pi) = 4(|\nu_0|, |\nu_\pi|)$ [62]. Following the analysis in the main text, $(\nu_0, \nu_\pi)$ could also determine the numbers of fractional-quasienergy corner modes in the $q$th-root descendants of the parent model $\mathcal{U}$ so long as the chiral symmetry $\Gamma$ is preserved.

## C  Stability to disorder

In this section, we demonstrate the stability of $q$th-root Floquet topological phases to disorder through numerical calculations. For the non-Hermitian FTI, we add random intracell coupling terms $H_{1d} = \sum_n W_n|n\rangle\langle n|\otimes\sigma_y$ to $H_1$ and $H_{2d} = \sum_n W_n'|n\rangle\langle n|\otimes\sigma_x$ to $H_2$ in Eqs. (19) and (20), respectively. Here $W_n, W_n'$ take different values for different unit cells $n$ and vary randomly in the range of $[-W, W]$. The form of disorder terms $H_{1d}$ and $H_{2d}$ are chosen to be general enough and also to ensure that the chiral symmetries of the parent and the $q$th-root systems

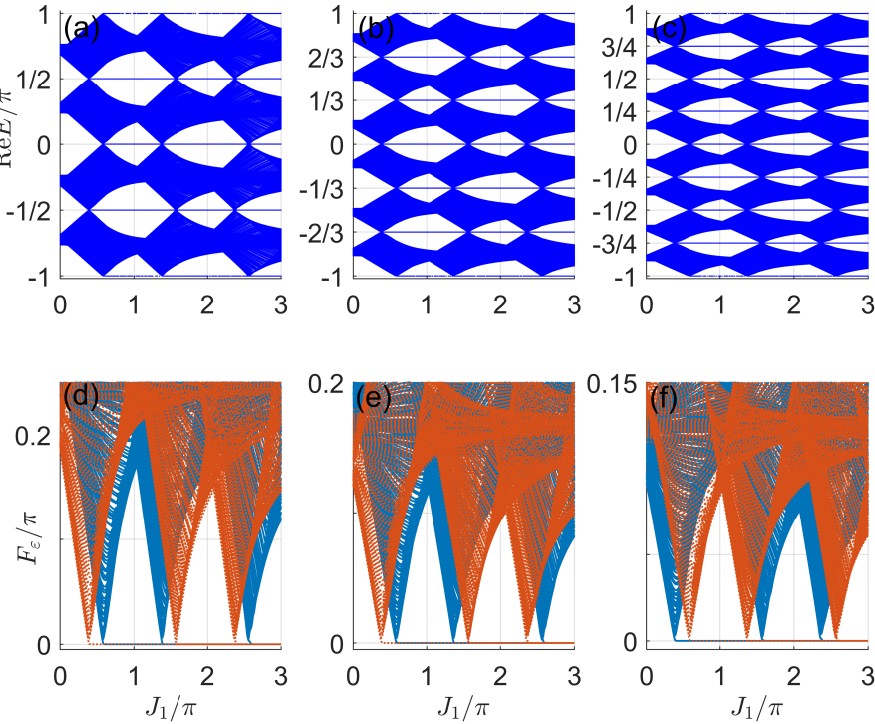

Figure 8: Floquet spectrum $E$ [(a)–(c)] and gap function $F_\varepsilon$ [(d)–(f)] of $U_{1/2}$ [Eq. (25)], $U_{1/3}$ [Eq. (26)] and $U_{1/4}$ [obtained from Eqs. (25), (6) and (7)] versus $J_1$ with disorder and under the OBC. The size of lattice, other system parameters and color schemes used for all panels are the same as those used for Figs. 2 and 3.

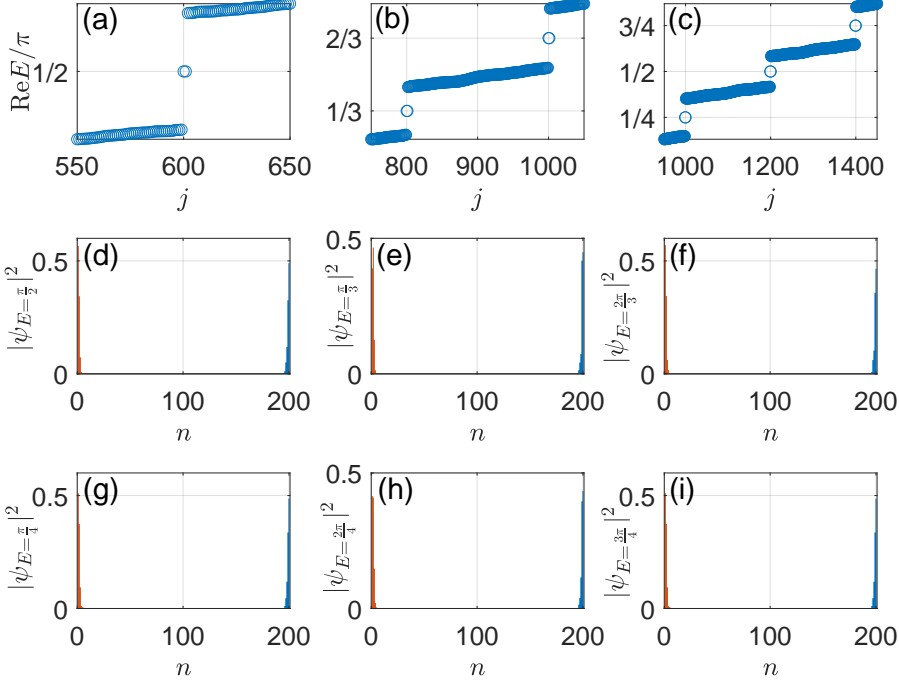

Figure 9: The real part of Floquet spectrum and fractional-quasienergy edge modes of the $q$th-root non-Hermitian FTIs for $q = 2, 3, 4$ with disorder. The notations, length of lattice and system parameters are the same as those used in Fig. 4.

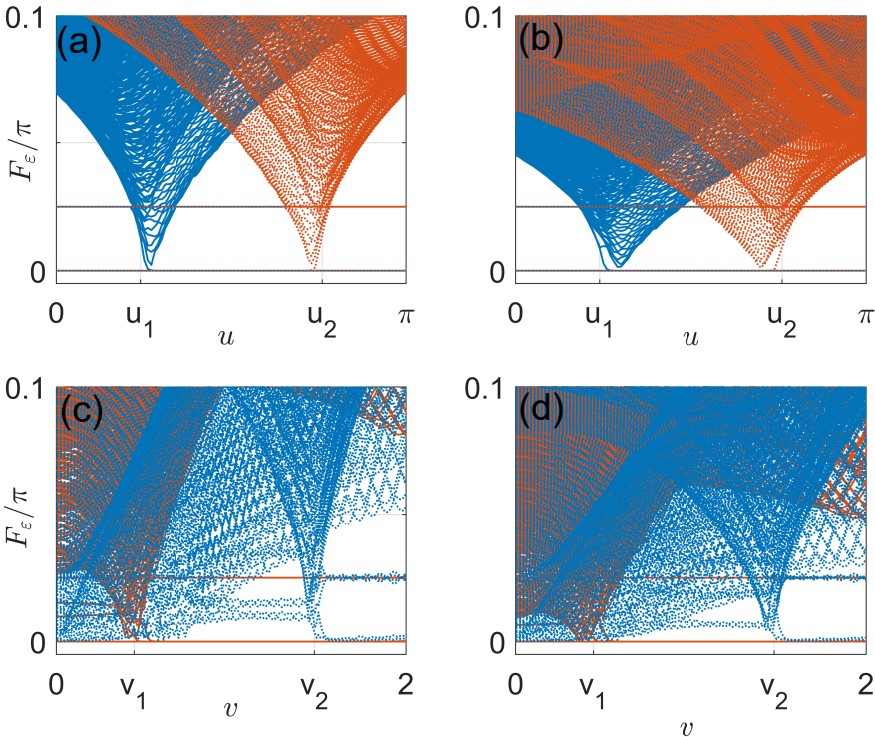

Figure 10: Gap function $F_\varepsilon$ of $\mathcal{U}_{1/2}$ [Eq. (31)] in (a), (c) and $\mathcal{U}_{1/3}$ [Eq. (32)] in (b), (d) versus $u$ and $v$ with disorder. The size of lattice and other system parameters are the same as those used in Figs. 5 and 6.

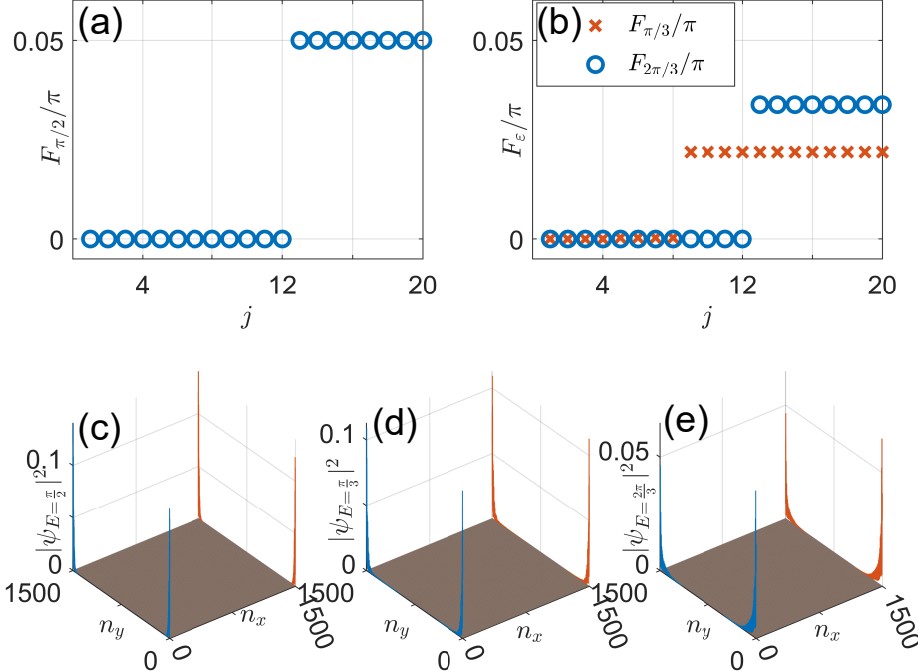

Figure 11: Gap functions and probability distributions of corner modes for $\mathcal{U}_{1/2}$ in (a), (c) and for $\mathcal{U}_{1/3}$ in (b), (d), (e). The notations, system parameters and size of lattice are the same as those used in Fig. 7. (c) shows four out of the twelve $\pi/2$ corner modes. (d) shows four out of the eight $\pi/3$ corner modes. (e) shows four out of the twelve $2\pi/3$ corner modes.

are preserved. In Fig. 8, we show the quasienergy spectrum and gap functions of the $q$th-root non-Hermitian FTI for $q = 2, 3, 4$ with the disorder amplitude $W = 0.2$ (comparable with the minimal energy scale of the clean system). The results show that the degenerate edge modes at different fractional quasienergies in the bulk spectrum gaps are well preserved under the impact of disorder. In Fig. 9, we further show the $E = \pi/2, \pi/3, 2\pi/3, \pi/4, 2\pi/4, 3\pi/4$ edge modes and their spatial profiles for $q = 2, 3, 4$ with the same disorder amplitude $W = 0.2$. It is clear that these fractional quasienergy edge modes indeed survive in the disordered system and are well localized around the sample boundary. The results presented in Figs. 8 and 9 are obtained for one disorder realization. We also checked a number of other disorder realizations in the calculation and found no observable difference.

For the 2D model, we introduce disorder by adding $\mathcal{H}_{1d} = \mathbb{I}_x \otimes \sum_n \mathcal{W}_n |n\rangle\langle n| \otimes \sigma_z$ and $\mathcal{H}_{2d} = \mathbb{I}_x \otimes \sum_n \mathcal{W}'_n |n\rangle\langle n| \otimes \sigma_x$ to $\mathcal{H}_{y1}$ and $\mathcal{H}_{y2}$ in Eqs. (23) and (24), respectively. The $\mathcal{W}_n$ and $\mathcal{W}'_n$ take different values for different cell indices $n$ and vary randomly in the range of $[-\mathcal{W}, \mathcal{W}]$. We also choose the disorder terms in $\mathcal{H}_{1d}$ and $\mathcal{H}_{2d}$ to be general enough and to make sure that the chiral symmetries of the parent and the $q$th-root models are retained. In Figs. 10 and 11, we present the gap functions and the spatial profiles of fractional-quasienergy corner modes of the $q$th-root non-Hermitian FSOTIs for $q = 2, 3$ and the disorder amplitude $\mathcal{W} = 0.2$ (comparable with the minimal energy scale of the clean system). The numerical results clearly suggest that the $q$th-root second order topological phases and their accompanying corner states in our system are robust to perturbations induced by symmetry-preserving disorder. The results presented in Figs. 10 and 11 are obtained for one disorder realization. We also checked a number of other disorder realizations in the calculation and found no observable difference.

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
