# Peer review of "$q$th-root non-Hermitian Floquet topological insulators"

_SciPost Physics, doi:SciPost Phys. 13, 015 (2022)_

## Round 1 · Referee Report · A.M. Marques · 2022-5-20

Strengths
1- Originality of the method to generate qth-root Floquet topological insulators.
2- The method covers any arbitrary qth-root, with q a positive integer.
3- These models allow for a controlled proliferation of topological edge states at any rational fractional quasienergies.
Weaknesses
1- Nothing major. Perhaps the construction of the qth-root generating procedure could be introduced in a more pedagogical way when it first appears in the paper, as I indicate below. This can be easily addressed by the authors.
Report
The authors developed a method for constructing qth-root versions of Floquet topological insulators (FTIs), generalizing the previous work of Ref. [45], which was limited to $2^n$th-root systems. In particular, the method allows one to generate topological states with any rational fractional quasienergy (in units of $\pi$), which is a key novelty of these models. Ultimately, the qth-root Floquet operator results from the multiplication of a diagonal matrix of U’s with a generalized shift operator, such that a block diagonal matrix is obtained when raising this operator to the qth power, with the blocks related to the parent model from which the topological properties are inherited. By taking the models of Refs.[62,66] as the parent models, several examples of qth-root FTIs are presented. I find these results novel and insightful, as well as relevant, given the recent surge in interest in high-root topology. Therefore, I recommend publication of this paper, provided the comments below are properly addressed first.
Requested changes
(1) I don’t know the current status of Ref. [45]. If it was already submitted, I believe the publication of this paper should only occur after the publication of Ref. [45], since this paper is a generalization of it.
(2) The general method outlined in eqs. (9)-(13) is not obvious at a first reading. Only after one explicitly works out an example can one understand it. I recommend the authors contextualize better this part by explaining in a couple of sentences, e.g. below eq. (10), the insights for introducing eqs. (11) and (12) such that eq. (13) is obtained. Also, immediately below eq. (13) it would be helpful if the authors worked out a specific example (say q=3), showing the form of the different $P, \tilde{H}, U$, etc., of the qth-root model. This would greatly clarify the method for the reader.
(3) The issue of the protecting symmetry of the edge states, discussed in the paragraphs above Sec. 3 and other places throughout, is a delicate point. The authors invariably relate it to the chiral symmetry of the parent model, which is true in a sense. However, at the level of the qth-root model, I would say that the 0 and $\pi$ edge states are protected by its chiral symmetry (which is built from copies of the $C$ symmetry of the parent model), while the fractional quasienergy edge states are not, since they are paired in symmetric quasienergies. For instance, in the square-root version of the SSH model we've studied in PRB 100, 041104(R) (2019), each of the finite energy edge states was shown to be protected by a “subchiral” symmetry $C_{1/2}$ (which incorporates the $C$ symmetry of the SSH model in one of its blocks, but not in the other), rather than by the global chiral symmetry $C_1$. I would expect something similar here, namely that each fractional quasienergy edge state should be protected by its own chiral-type symmetry, which locks that state at its specific gap. In other words, the protecting symmetries of the qth-root model, while closely connected to the parent model, should be derived directly from the starting qth-root model. I would like to hear the authors’ views on this, and eventually the inclusion of some extra clarifications on these points above Sec. 3.
(4) In the penultimate paragraph of Sec. 4, the authors classify the extra 1D edge states appearing in Figs. 4 and 5 as “trivial edge states”. Are the authors sure of this? Since, for instance, the parent model consists of dimerized SSH chains in the x direction and in the topological phase, I would expect these 1D states to be topological edge states coming from the weak topology of the system (as is known to occur, e.g., in the 2D SSH model).
Minor typos/comments on technical details:
i) I could be mistaken, but there is a general feeling that the $U_1$ and $U_2$ order, as defined in eq. (1), is not respected everywhere in the calculations below it. E. g., substituting (1) in (3) leads me to (4) with tau_+ and tau_- switched. Please recheck these calculations.
ii) In eq. (12), check if the$P_j\to P_{q-j}$ (with $P_q=P_0$) substitution should be made for the diagonal terms to have the correct order $U_1,U_2,U_3$,…
iii) In eq. (11) and below eq. (13), correct the index of the product.
iv) Below eq. (15): “characterized integer” $\to$ “characterized by integer”
v) Above eq. (20), the authors identify eq. (14) with $H(t)$ and eq. (15) with $H(t+1/2)$. In connection with point i), check if it should not be the converse instead, given the $U_1$ and $U_2$ definition in eq. (1). The same above eq. (26).
vi) Above (21) and again above eq. (27), $H_1$ and $H_2$ are switched.
vii) In the second paragraph below eq. (21): “generate states with E=±0,±2π/3 (E=±π/3)” $\to$ “generate states with E=±0,±2π/3 (E=±π/3,±π)”.
viii) Sec. A, 5th line from the end: the second $ν_0$ should be $ν_π$.

---

## Round 1 · Referee Report · Anonymous · 2022-6-1

Strengths
1 Clear presentation
2 Convenient notation transferable to different fields in which floquet phases are discussed
3 original results regarding the conceptualization of exotic non-Hermitian topological states
4 possible realization of Z3 parafermions
Weaknesses
1 Too technical
2 No feasiblity / stability analysis of the proposed topological states
Report
The manuscript discusses an nth root extension of non-Hermitian topological states through Flouqet engineering. Some of the results do explicitly build up on previous work, but there are also original aspects in the manuscript. In principle I appreciate this work and tend to recommend it for publication.

---

## Round 2 · Referee Report · A.M. Marques (Referee 1) · 2022-6-17

Report

I believe the authors have properly addressed the issues raised by both referees, which led to the inclusion of several new paragraphs and a new appendix. In particular, I greatly enjoyed reading the additional results pertaining to the generalized “subchiral” symmetry that the authors defined, whose combination with the global chiral symmetry, together with the mod 2$\pi$ constraint on the quasienergies, leads to relevant new results regarding the topological protection. As such, I recommend publication of the manuscript in its current form.

Requested changes

Misprints:
• The authors missed the “$j\to l$” index substitution in the product below eq. (13)
• Below eq. (18): “spectral and topology” -> “spectral and topological properties”

---

## Round 2 · Author Response

Editor
SciPost Physics

Dear Editor,

Thank you so much for facilitating the peer review of our manuscript entitled “qth-root non-Hermitian Floquet topological insulators”. We are happy that both referees have provided encouraging comments and supported the publication of our manuscript in SciPost Physics. We also appreciate the comprehensive questions and suggestions raised by our Referee 1 (A. M. Marques) that have been very helpful in further improving the quality of our manuscript.

Please find below a list of our point-to-point responses to the referees’ comments. We have also made appropriate changes in the revised manuscript. These are highlighted in red for the convenience of the referees. We look forward to hearing from you again soon.

Yours truly,
Longwen Zhou, Raditya Weda Bomantara, and Shenlin Wu

Report 1:

Weakness: Nothing major. Perhaps the construction of the qth-root generating procedure could be introduced in a more pedagogical way when it first appears in the paper, as I indicate below. This can be easily addressed by the authors.

Reply:
We thank our referee for suggesting a way to improve our manuscript. We have carefully followed the referee’s suggestions below to make the presentation of our qth-root system’s construction more accessible to general readers.

Requested changes:
(1) I don’t know the current status of Ref. [45]. If it was already submitted, I believe the publication of this paper should only occur after the publication of Ref. [45], since this paper is a generalization of it.

Reply:
We thank our First Referee for this comment. Actually, the Ref. [45] was submitted to PRL long before the submission of this article to SciPost Physics. At present, it is still under the second round of review in PRL. While it would indeed be ideal to have this manuscript published after the Ref. [45], the actual manuscript handling process is however beyond our control. Therefore, in our humble opinion, strictly waiting for the publication of Ref. [45] may not be the best course of action, since it is then possible that other articles that are built on some aspects of this work, which may not be written us, may similarly need to wait for this article to be published. Fortunately, the presence of arXiv makes it possible for the Ref. [45] to be properly recognized even while it is still under review. In this case, it should then still be okay for the current article to be processed independently of the Ref. [45].

(2) The general method outlined in eqs. (9)-(13) is not obvious at a first reading. Only after one explicitly works out an example can one understand it. I recommend the authors contextualize better this part by explaining in a couple of sentences, e.g. below eq. (10), the insights for introducing eqs. (11) and (12) such that eq. (13) is obtained. Also, immediately below eq. (13) it would be helpful if the authors worked out a specific example (say q=3), showing the form of the different P, H, U, etc., of the qth-root model. This would greatly clarify the method for the reader.

Reply:
We thank our First Referee for the invaluable suggestion that will indeed improve the clarity of our manuscript. Following this suggestion, in the revised manuscript, we have now provided the intuitive mechanism behind our construction of Eq. (12), as well as its application to a specific example of q=3. A new figure (Fig. 1 in the revised manuscript) has also been added to further enhance our exposition.

(3) The issue of the protecting symmetry of the edge states, discussed in the paragraphs above Sec. 3 and other places throughout, is a delicate point. The authors invariably relate it to the chiral symmetry of the parent model, which is true in a sense. However, at the level of the qth-root model, I would say that the 0 and π edge states are protected by its chiral symmetry (which is built from copies of the C symmetry of the parent model), while the fractional quasienergy edge states are not, since they are paired in symmetric quasienergies. For instance, in the square-root version of the SSH model we've studied in PRB 100, 041104(R) (2019), each of the finite energy edge states was shown to be protected by a “subchiral” symmetry C1/2 (which incorporates the C symmetry of the SSH model in one of its blocks, but not in the other), rather than by the global chiral symmetry C1. I would expect something similar here, namely that each fractional quasienergy edge state should be protected by its own chiral-type symmetry, which locks that state at its specific gap. In other words, the protecting symmetries of the qth-root model, while closely connected to the parent model, should be derived directly from the starting qth-root model. I would like to hear the authors’ views on this, and eventually the inclusion of some extra clarifications on these points above Sec. 3.

Reply:
We sincerely thank our First Referee for the comment about the symmetry protection of fractional edge/corner states, and especially also for pointing out the referee’s previous paper (now cited as Ref. [79] in the revised manuscript) that has helped us tremendously in addressing the comment. As the referee suspected, there is indeed an additional chiral-like symmetry that acts only within the subspace of the enlarged degree of freedom, roughly similar to what the referee discovered in Ref. [79]. In particular, for the case of square-root Floquet topological phases, we identify the presence of C_(1/2) symmetry which acts on the square-root Floquet operator as C_(1/2) U_(1/2) 〖C_(1/2)〗^†=-U_(1/2). As a result, the quasienergies of U_(1/2) come in pairs of E' and E'±π. In combination with the usual chiral symmetry C, quasienergies which satisfy E'±π=-E', i.e., E'=±π/2, are at least two-fold degenerate, whose associated quasienergy eigenstates can be chosen to simultaneously be ±1 eigenstates of the product CC_(1/2). In the context of edge/corner states, the ±1 eigenstates of CC_(1/2) correspond to states localized at two opposite edges/corners. Due to the discreteness of CC_(1/2) eigenvalues, such states remain pinned at quasienergies ±π/2 even in the presence of symmetry-preserving perturbations.
The above argument can be extended to the more general qth-root setting. In this case, we can similarly identify a generalized “subchiral” symmetry C_(1/q) which operates as C_(1/q) U_(1/q) 〖C_(1/q)〗^†=ω^† U_(1/q), where ω=□exp(2iπ/q), forcing the quasienergies of U_(1/2) to cluster as E'-2πj/q for j=0,1,⋯q-1. Following the same argument as above, the product of the usual chiral symmetry and such a subchiral symmetry then protects the fractional quasienergy edge/corner states of our qth-root systems.
Following the referee’s suggestion, we have also implemented the above discussion in the revised manuscript, just before Sec. 3.

(4) In the penultimate paragraph of Sec. 4, the authors classify the extra 1D edge states appearing in Figs. 4 and 5 as “trivial edge states”. Are the authors sure of this? Since, for instance, the parent model consists of dimerized SSH chains in the x direction and in the topological phase, I would expect these 1D states to be topological edge states coming from the weak topology of the system (as is known to occur, e.g., in the 2D SSH model).

Reply:
We thank our First Referee for raising this interesting question. The “trivial edge states” with quasienergies away the fractional quasienergy corner modes are formed by coupling the bulk states along x-direction (corresponding to the bulk states of SSH chain) and the edge states along y-direction of the lattice. As the 1D chain along y-direction also possesses the chiral symmetry, the coupling of its degenerate edge modes with the bulk states along x-direction can yield degenerate states in 2D that are protected by the chiral symmetry of a 1D subsystem. In this sense, we may indeed regard the “trivial edge states” mentioned in our manuscript as “weak edge states”. In response to the suggestion of our First Referee, we have mentioned this point and give more discussions about these “weak edge states” in our revised manuscript.
In the meantime, we would like to emphasize that such weak edge states are not topological compared with the corner modes because their number is sensitive to the system size along x-direction and their quasienergies are sensitive to the choice of system parameters. The fact that the quasienergies of these “weak edge states” look the same in certain ranges of system parameters is due to our choice of a flat-band limit for the SSH chains along the x-direction, so that all the bulk states of the SSH chains have the same quasienergies (e.g., E=pi/20 around the zero corner modes). Away from this flat-band limit, different bulk states of the SSH chains will have different quasienergies. The 2D states formed by coupling these dispersive SSH bulk states with the edge states of the lattice along y-direction will then also possess different quasienergies instead of being degenerate at the original values of E.

Minor typos/comments on technical details:

i) I could be mistaken, but there is a general feeling that the U1 and U2 order, as defined in eq. (1), is not respected everywhere in the calculations below it. E. g., substituting (1) in (3) leads me to (4) with tau_+ and tau_- switched. Please recheck these calculations.

Reply:
We thank our First Referee very much for raising this issue. We have rechecked our derivations and found a typo in the range of integration in Eq. (3). We have fixed this issue and the original Eq. (4) looks okay after this correction. We have also fixed related issues in other parts of the manuscript.

ii) In eq. (12), check if the Pj→Pq−j (with Pq=P0) substitution should be made for the diagonal terms to have the correct order U1, U2, U3,…

Reply:
In the revised manuscript, we have corrected the equation following the suggestion of our First Referee. Specifically, by simply changing the definition of \tilde{H} according to the revised Eq. (11), Eq. (13) remains consistent as before.

iii) In eq. (11) and below eq. (13), correct the index of the product.

Reply:
We have corrected the index following the suggestion of our Referee.

iv) Below eq. (15): “characterized integer” → “characterized by integer”

Reply:
We have corrected this typo following the suggestion of our Referee.

v) Above eq. (20), the authors identify eq. (14) with H(t) and eq. (15) with H(t+1/2). In connection with point i), check if it should not be the converse instead, given the U1 and U2 definition in eq. (1). The same above eq. (26).

Reply:
We have revised the discussions around Eqs. (20) and (26) following the suggestion of our Referee. They are now consistent with the conventions used in the Sec. 2 of our revised manuscript.

vi) Above (21) and again above eq. (27), H1 and H2 are switched.

Reply:
In the revised manuscript, we have added an explicit demonstration for the construction of U_{1/3} in Sec. 2. The descriptions above Eqs. (21) and (27) in the previous manuscript have been revised accordingly.

vii) In the second paragraph below eq. (21): “generate states with E=±0,±2π/3 (E=±π/3)” → “generate states with E=±0,±2π/3 (E=±π/3,±π)”.

Reply:
We have added the missing information ±π in our revised manuscript.

viii) Sec. A, 5th line from the end: the second ν0 should be νπ.

Reply:
We have corrected this typo in the revised manuscript.

Report 2:

Weakness:
1 Too technical

Reply:
In response to the weakness raised by our Second Referee, we have presented the intuition behind our qth-root construction in the revised Sec. II. Moreover, to demonstrate the application of our construction, we have also added an explicit construction of the cubic-root model and an illustrative graph in the Sec. II of our revised manuscript. We hope that these additions could make the physical picture and theoretical framework presented in our work more transparent.

2 No feasibility / stability analysis of the proposed topological states

Reply:
In response to the weakness raised by our Second Referee, we have discussed the robustness of topological edge/corner states to disorder in a new Appendix C of our revised manuscript. With the help of numerical calculations, we showed that the edge/corner states in our rooted Floquet systems are stable under perturbations induced by disorder that does not break the protecting symmetries of the Floquet system. We have also discussed possible experimental setups that could be used to detect the proposed topological states in the conclusion section of the manuscript.

Report:
The manuscript discusses an nth root extension of non-Hermitian topological states through Flouqet engineering. Some of the results do explicitly build up on previous work, but there are also original aspects in the manuscript. In principle I appreciate this work and tend to recommend it for publication.

Reply:
We thank our Second Referee for recommending our work to publication. In the revised manuscript, we have tried our best to address the weaknesses raised by our Second Referee.

We benefited a lot from the above exchanges and we thank once again our Referees for the very constructive comments and suggestions, which helped us greatly in improving the quality of our manuscript.

---

## Round 2 · List of Changes

1. Corrected typos in Eqs. (3), (7), (11).
  2. A new added Figure 1 to illustrate the generation cubic-root systems in Sec. 2.
  3. Explicit discussions for the generation cubic-root systems in Sec. 2.
  4. Discussion of the protecting symmetries for the fractional quasienergy edge/corner modes in Sec. 2.
  5. Corrected typos in the words above and below Eqs. (25) and (31).
  6. Discussion of the weak edge states before the last paragraph of Sec. 4.
  7. Add a new Appendix C to discuss the stability of qth-root non-Hermitian Floquet topological phases to disorder.

---

## Editorial Decision

published